# Hypoxia promotes tumor immune evasion by suppressing MHC-I expression and antigen presentation

Hala Estephan [ID][1], Arun Tailor [ID][2], Robert Parker[2], McKenzie Kreamer[3], Ioanna Papandreou[3], Leticia Campo [ID][1], Alistair Easton[1], Eui Jung Moon [ID][1], Nicholas C Denko[3], Nicola Ternette [ID][2], Ester M Hammond [ID][1] & Amato J Giaccia [ID][1][✉]

## Abstract

**Hypoxia is a common feature of solid tumors that has previously been linked to resistance to radiotherapy and chemotherapy, and more recently to immunotherapy. In particular, hypoxic tumors exclude T cells and inhibit their activity, suggesting that tumor cells acquire a mechanism to evade T-cell recognition and killing. Our analysis of hypoxic tumors indicates that hypoxia downregulates the expression of MHC class I and its bound peptides (i.e., the immunopeptidome). Hypoxia decreases MHC-I expression in an oxygen-dependent manner, via activation of autophagy through the PERK arm of the unfolded protein response. Using an immunopeptidomics-based LC-MS approach, we find a significant reduction of presented antigens under hypoxia. Inhibition of autophagy under hypoxia enhances antigen presentation. In experimental tumors, reducing mitochondrial metabolism through a respiratory complex-I inhibitor increases tumor oxygenation, as well as MHC-I levels and the immunopeptidome. These data explain the molecular basis of tumor immune evasion in hypoxic conditions, and have implications for future therapeutic interventions targeting hypoxia-induced alterations in antigen presentation.**

**Keywords** Hypoxia; MHC I Expression; Antigen Presentation; Autophagy; Immune Response
**Subject Categories** Autophagy & Cell Death; Cancer; Immunology

## Introduction

The microenvironment within most solid tumors is characterized by regions of low oxygen, known as hypoxia. Hypoxia plays a pivotal role in tumorigenesis and has been associated with several mechanisms of resistance to commonly used therapies including immunotherapy (Pietrobon and Marincola, 2021; Singleton et al, 2021). Mechanisms of immune evasion include the alteration of cytokine expression, the recruitment of immunosuppressive cell populations and the expression of co-inhibitory ligands such as PD-L1 that regulate T-cell activity (Barsoum et al, 2014; Noman et al, 2015; Wu et al, 2022). Modification of MHC I expression is a critical component of immune evasion (Roh et al, 2017; Wu et al, 2023). However, the role of hypoxia in the regulation of MHC I expression and the subsequent impact on tumor immunogenicity and CD8[+]T cells response has not been well defined.

MHC I proteins are critical mediators of the adaptive immune response, responsible for presenting thousands of peptide antigens (8–12 amino acids) on the cell surface. Peptides presented by MHC I are derived from the proteolysis of intracellular proteins giving rise to a diverse array of peptide MHC I complexes known collectively as the immunopeptidome (Yewdell, 2022; Kubiniok et al, 2022; Dersh et al, 2021). Foreign antigens belonging to viruses or bacteria and mutated antigens derived from somatic changes presented by MHC I are recognized by CD8[+]T cells, leading to the activation of the immune response to eliminate cells expressing these antigens (Hu et al, 2018; Xie et al, 2023; Jou et al, 2021). However, antigen presentation is a highly dynamic process continuously sampling the internal proteome of a cell, and therefore alterations in the cellular microenvironment influences the presentation of antigens on MHC I (Neal et al, 2018). Tumor cells employ various strategies to reduce antigen presentation on their surface, including antigen depletion, the genetic downregulation of MHC I expression, the modulation of transcription and post-transcriptional processes, ultimately resulting in depletion of the immunopeptidome (Jhunjhunwala et al, 2021). Recent developments in mass spectrometry enable monitoring of small changes in the immunopeptidome through the immunoaffinity selection of MHC complexes (Bassani-Sternberg and Coukos, 2016; Purcell et al, 2019). While the focus on understanding changes in the immunopeptidome of tumor cells has been based on genetic

[1]Department of Oncology, The University of Oxford, Oxford OX3 7DQ, UK. [2]Centre for Immuno-Oncology, Nuffield Department of Medicine, University of Oxford, Oxford OX37BN, UK. [3]Department of Radiation Oncology, OSU Wexner Medical Center, James Cancer Hospital and Solove Research Institute, Ohio State University, Columbus, OH, USA. ✉E-mail: amato.giaccia@oncology.ox.ac.uk

alterations, we hypothesized that the tumor microenvironment could also modulate the immunopeptidome, providing additional pathways for immune evasion. To date, the characterization of endogenously presented peptides under hypoxic conditions has not been rigorously explored.

Therefore, we investigated the relationship between hypoxia and antigen presentation using colorectal cancer cell lines. We found that hypoxia leads to a reduction in MHC I expression and subsequently a decrease in CD8+T-cell infiltration. The decrease of MHC I expression in an oxygen-dependent manner is the result of autophagy-targeted degradation that is signaling through the activation of the PERK arm of the unfolded protein response (UPR). Immunopeptidomics methodology revealed a significant decrease in the number of presented antigens under hypoxia, consistent with a reduction in MHC I. Interestingly, blocking PERK or autophagy under hypoxic conditions increased the MHC I expression and the number of peptides unveiling new antigens. Moreover, in vivo data revealed that pharmacological regulation of hypoxia by decreasing mitochondrial respiration increased MHC I and the immunopeptidome, suggesting a potential therapeutic strategy for use in a clinical setting.

# Results

## Hypoxia is linked to decreased CD8+T-cell infiltration in colorectal cancer

Hypoxia has been shown to impede the infiltration of CD8+T cells into tumors (Gajewski et al, 2013). To better understand the role of hypoxia in regulating CD8+T-cell infiltration, we first analyzed the CD8+T-cell distribution in colorectal cancer patient tissue microarray using multiplexed spectral fluorescence microscopy; hypoxia was determined by CAIX staining. Our data indicated that both hypoxia and CD8+T-cell distribution within the tissue was heterogeneous and most of these tumors presented low levels of CD8+T cells (Fig. 1A; Appendix Fig. S1A) which were even lower in CAIX-positive areas (Fig. 1B). The percentage of CD8+T cells correlated negatively with the percentage of cells positive for the hypoxia marker, CAIX (Appendix Fig. S1B). In addition, proximity analysis of different immune cells including CD8+, CD4+T cells and macrophages showed that the majority of CD8+T cells were significantly excluded from the hypoxic regions compared to the other immune cells (Fig. 1C,D; Appendix Fig. S1C–E). We then determined whether the exclusion of CD8+T cells could be reversed by modulating hypoxia in vivo using papaverine, an FDA-approved drug that has been shown to inhibit mitochondrial complex-I and oxygen consumption, thereby reducing hypoxia in tumors (Benej et al, 2018). In this experiment, mouse colorectal CT26 cancer cells were subcutaneously transplanted into syngeneic (Balb/c) mice, and papaverine was administered at a dose of 2 mg/kg every 24 h for 3 days before tumor was harvested (Fig. 1E). Strikingly, immuno-fluorescence staining of tumor sections showed an increase in the number of CD8+T cells in the papaverine-treated tumor compared to the vehicle-treated control tumor; GLUT1 staining, a HIF1 and hypoxia marker, further supported decreases in hypoxic regions in the papaverine-treated group (Fig. 1F,G). Together, these results demonstrate that hypoxia is a key factor contributing to the exclusion of CD8+T cells in human and mouse colorectal cancer.

## Hypoxia is associated with reduced levels of MHC I expression

To explore the hypoxia-induced changes in the colorectal cancer proteome profile, a label-free quantitative proteomics experiment was carried out in HT29 human colorectal cancer cell line exposed to normoxia or hypoxia for 24 h. Using the Ingenuity Pathway Analysis (IPA) software, we evaluated the changes in expressed proteins ($P > 0.05$, FC > 1.5). The top canonical pathways with a $-\log(P$ value) greater than 1.5 were graphically represented showing positive (orange) or negative (blue) z-scores according to upregulated or downregulated pathways, respectively (Fig. 2A). Interestingly, the MHC I mediated antigen processing and presentation pathway ($-\log P = 1.8$) was among the downregulated pathways. This is significant because the inhibition of MHC I expression impairs the presentation of antigens on tumors that would be recognized by CD8+T cells, leading to immune evasion (Taylor and Balko, 2022; Dhatchina-moorthy et al, 2021). Our next step was then to evaluate the role of hypoxia in directly regulating MHC I levels; we determined changes in MHC I expression in HT29 and HCT116 colorectal cancer cell lines that were exposed to a range of oxygen levels (<0.1–2% $O_2$) for 24 h. Flow cytometry-based analysis of the surface levels of MHC I showed that the expression of MHC I was reduced in an oxygen-dependent manner in both cell lines, and it was downregulated the most when cells were exposed to the lowest oxygen levels (Fig. 2B,C; Appendix Fig. S2A). These changes were also observed in other tumor types such as Calu6 and A549, human lung cancer cells exposed to <0.1% $O_2$ (Appendix Fig. S2B). In addition, MHC I protein levels were reduced in a time-dependent manner in HT29 and HCT116 cells exposed to hypoxia with significant reductions in MHC I protein levels seen within 8 h of exposure (Fig. 2D; Appendix Fig. S2C). These data were further supported by immunofluorescence staining showing a reduction of MHC I expression on the cell surface (Appendix Fig. S2D,E). Interestingly, gene expression analysis demonstrated a significant increase in the mRNA levels of MHC I subtypes in HT29 (*HLA*A1, HLA*C4)* and HCT116 cells (*HLA*A1, HLA*A2)* treated under hypoxia suggesting that the hypoxia-mediated reduction of MHC I expression occurs post-transcriptionally (Fig. 2E; Appendix Fig. S2F). Hypoxia-mediated repression of MHC I expression was further confirmed in a colorectal cancer patient tissue microarray using multiplexed spectral fluorescence microscopy. Hypoxic regions, represented by CAIX staining, had a significant negative correlation with MHC I-positive cells (red) (Fig. 2F,G). Immunofluorescence staining of mouse tumor sections treated with papaverine showed a decrease in the expression of GLUT1 while increasing MHC I expression, suggesting that reduced hypoxia by papaverine elevates levels of MHC I (Fig. 2H–J). Hypoxia-inducible factor (HIF) is a master regulator of cellular responses under hypoxia (LaGory and Giaccia, 2016), to examine whether HIF plays a role in the alteration of MHC I expression, we exposed RKO cells lacking HIF1α to hypoxia and showed a reduction in total MHC I protein levels compared to the control. Similarly, MHC I expression was reduced after knocking down ARNT/HIF1β in HT29 cells, indicating that the decrease in MHC I expression is HIF-independent (Appendix Fig. S2G–I). Taken together, these data show that MHC I expression is reduced in an oxygen-dependent manner under hypoxia and that this contributes to the decrease in CD8+T-cell infiltration.

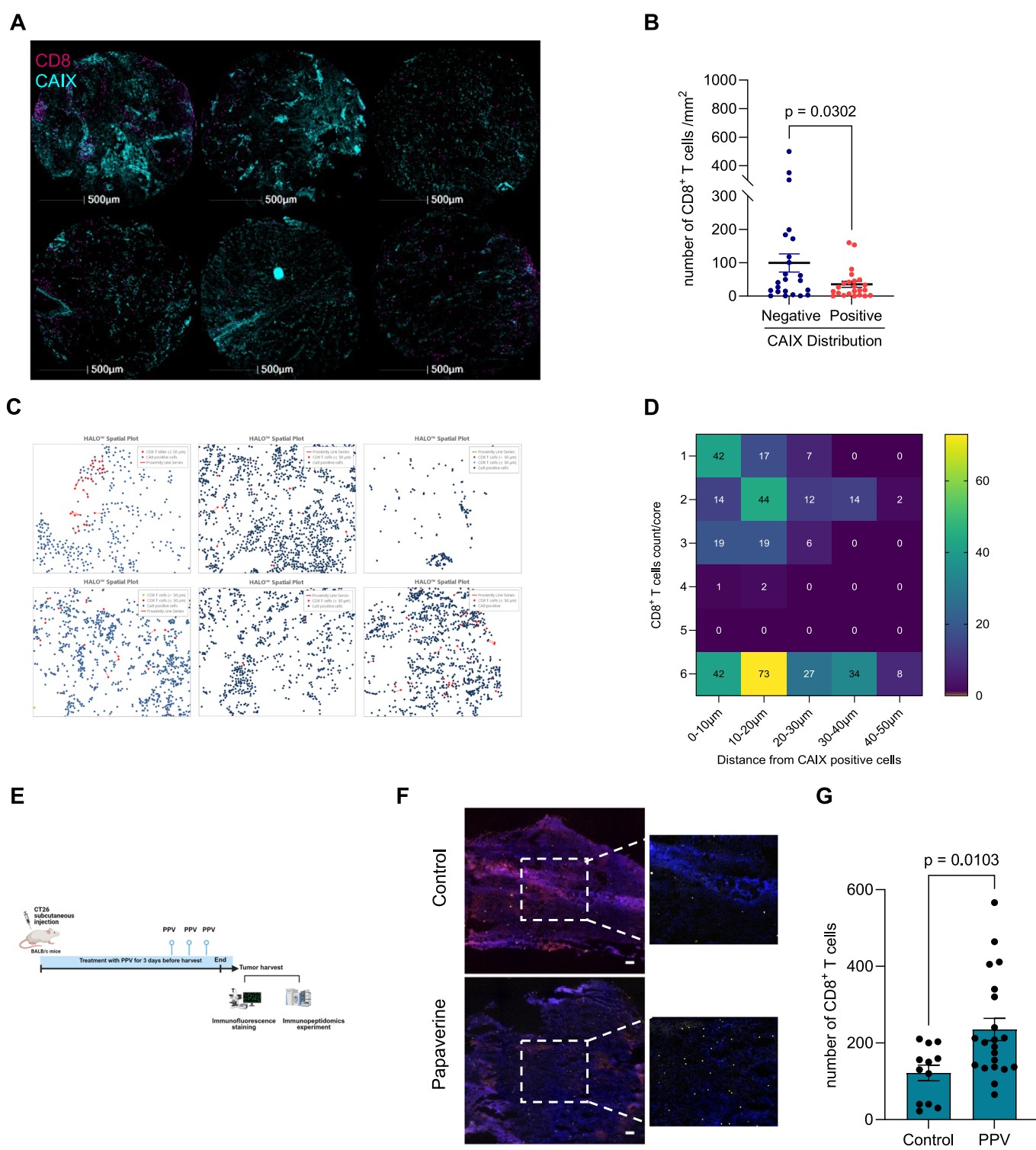

## Unfolded protein response-dependent MHC I expression under hypoxia

While we did not find a role for the HIF pathway in regulating MHC I, pathway analysis (Fig. 2A) and western blot showed robust activation of the unfolded protein response (UPR) (indicated by PERK, GRP78, IREα and ATF4) in HT29 cells when subjected to hypoxia (Fig. 3A). Activation of the UPR pathways regulate cellular homeostasis in response to severe hypoxic conditions (Wouters and Koritzinsky, 2008). Considering that downregulation of MHC I was observed under extreme hypoxia (<0.1% O$_2$), we next determined whether hypoxia-mediated MHC I reduction was signaled by any of

**Figure 1. Hypoxia is linked to decreased CD8+T-cell infiltration in colorectal cancer.**

(A) Multiplexed spectral microscopy imaging showing the distribution of CAIX-positive cells (green) and CD8+T cells (magenta) in 6 different tissue microarrays from patients with colorectal cancer (scale bar: 500 μm). (B) Graph showing the distribution of CD8+T cells between CAIX-negative and -positive areas. Data represent mean ± SEM from 23 different tissue microarrays from patients with colorectal cancer. Statistical analysis was calculated using paired Student's $t$ test. (C) Spatial plot from Halo analysis showing the proximity of CD8+T cells within 50 μM from CAIX-positive cells. (D) Heatmap showing the number of CD8+T cells distribution into the tissue based on the distance from CAIX-positive areas. (E) Schematic overview showing the in vivo study plan, this scheme was created with biorender.com. Papaverine PPV. (F) Immunofluorescence staining of a representative tumor tissue of control and papaverine depicting CD8+T cells (yellow), GLUT1 (red) and DAPI (blue) staining (scale bar: 200 μm). (G) Quantification of the number of CD8+T cells in different tumors sections in control and papaverine conditions. Data represents mean ± SEM from at least four sections from three different mice. Statistical significance was determined with unpaired Student's $t$ test. Source data are available online for this figure.

the three arms of the UPR, through pharmacological inhibition. We found that inhibition of the PERK arm using AMG PERK 44 increased the total levels of MHC I in HT29 and HCT116 cells (Fig. 3B; Appendix Fig. S3A). Furthermore, inhibition of PERK during hypoxic exposure resulted in an increase in surface levels of MHC I expression in both cell lines (Fig. 3C; Appendix Fig. S3B). Likewise, immunofluorescence staining for MHC I in HT29 cells following exposure to hypoxia revealed a decrease in MHC I expression and the reduction was reversed by PERK inhibition (Fig. 3D,E). In contrast, MHC I levels were not rescued when ATF6 and IREα were inhibited under hypoxia (Appendix Fig. S3C). To further support a role for the UPR in regulating MHC I, we treated cells with Thapsigargin and Tunicamycin, both well-characterized activators of the UPR (Sehgal et al, 2017; Bull and Thiede, 2012) and observed a significant reduction in the surface expression of MHC I in both HT29 and HCT116 cells (Fig. 3F,G; Appendix Fig. S3D,E). While MHC I protein levels decreased, gene expression analysis demonstrated a significant increase in mRNA levels of MHC I subtypes in HT29 (*HLA*A1, HLA*C4*) treated with Thapsigargin and Tunicamycin, confirming that the reduction of MHC I expression by PERK occurs at the protein level (Appendix Fig. S3F,G). To understand the mechanism of degradation of MHC I, we investigated the role of the cell membrane proteins, SUSD6 and TMEM127, which form a tripartite complex with MHC I and act as negative regulators (Chen et al, 2023). Interestingly, exposure of HT29 cells to hypoxia increased the expression of both SUSD6 and TMEM127 at the mRNA level (Appendix Fig. S4A,B) and the protein level (Fig. 3H,I; Appendix Fig. S4C), indicating that SUSD6 and TMEM127 are hypoxia-inducible genes. To further determine whether these proteins were regulated by the UPR pathway, we treated HT29 cells with Thapsigargin or Tunicamycin and found a significant increase in SUSD6 and TMEM127 mRNA levels (Appendix Fig. S4D). Moreover, both SUSD6 and TMEM127 expression was decreased under hypoxia when HT29 cells were treated with PERK inhibitor (AMG44) (Appendix Fig. S4E); using integrative genomics viewer (IGV) browser we found that SUSD6 and TMEM127 genes has multiple binding sites for the transcription factor ATF4 (Appendix Fig. S4F) suggesting that these proteins are direct downstream targets of the UPR. Most importantly, knocking down SUSD6 significantly increased plasma membrane level of MHC I in HT29 cells, revealing a role of this protein in regulating MHC I in hypoxia (Fig. 3J; Appendix Fig. S4G). In contrast, knocking down TMEM127 did not substantially increase the levels of MHC I in HT29 cells (Appendix Fig. S4H,I). In addition, our proteomics analysis identified the E3 ligase STUB1, which has recently been recognized as a negative regulator of MHC I expression (Chen et al, 2023). We investigated the role of STUB1

in regulating MHC I expression under hypoxic conditions and found a significant positive correlation between STUB1 mRNA and a validated hypoxic signature, indicating that STUB1 mRNA expression is induced in hypoxic tumors (Appendix Fig. S4J). Inhibition of STUB1 increased the plasma membrane level of MHC I in HT29 cells, supporting its role in regulating MHC I in hypoxia (Appendix Fig. S4K). Taken together, these data provide strong evidence for the involvement of UPR-mediated PERK signaling in the degradation of MHC I under hypoxia through SUSD6 and STUB1.

## Activation of autophagy induces MHC I degradation under hypoxia

Since the activation of the UPR under hypoxia has been shown to have a role in promoting autophagy (Rouschop et al, 2010), we hypothesized that hypoxia induces PERK signaling that leads to the activation of autophagy and MHC I degradation. In exploring the proximal portion of this relationship, we found that autophagy increased in response to hypoxia in HT29 and HCT116 cells (Fig. 4A; Appendix Fig. S5A) and that inhibition of PERK resulted in an accumulation of p62, a known substrate for autophagy, indicating that autophagy is blocked (Fig. 4B). Since autophagy has been described to play a role in regulating MHC I expression (Gestal-Mato and Herhaus, 2023; Øynebråten, 2020), we further explored whether reduced levels of MHC I under hypoxia could be autophagy-dependent. To investigate this hypothesis, we blocked autophagy at the level of the lysosome using the V-ATPase inhibitor, bafilomycin A1 (Baf A1), and found that the reduced surface levels of MHC I in HT29 and HCT116 under hypoxic conditions were significantly recovered (Fig. 4C; Appendix Fig. S5B). In addition, immunofluorescence staining of HT29 cells treated with Baf A1 under hypoxic conditions confirmed a significant recovery of total MHC I expression and revealed an increase in intracellular expression suggesting that MHC I localized within autophagosomes and lysosomes when autophagy was inhibited (Fig. 4D,E). Colocalization of MHC I and LAMP1 staining further suggests that autophagy inhibition under hypoxia resulted in the accumulation of MHC I within the lysosomes (Fig. 4F,G; Appendix Fig. S5C). Similar results were found when HT29 cells when treated with autophagy inhibitor chloroquine, which inhibits autophagic flux by disrupting the fusion of the autophagosome and the lysosome (Appendix Fig. S5D). Genetic inhibition of two essential components of the autophagosome, ATG5 and ATG12, by siRNAs also increased the total and membrane-bound MHC I levels in HT29 cells (Appendix Fig. S5E–G). Therefore, our data supports a specific role for autophagy in the trafficking of MHC I to the lysosomes under hypoxia. Autophagy can selectively degrade target molecules using cargo receptor proteins that bind and recruit substrates to autophagosomal membranes

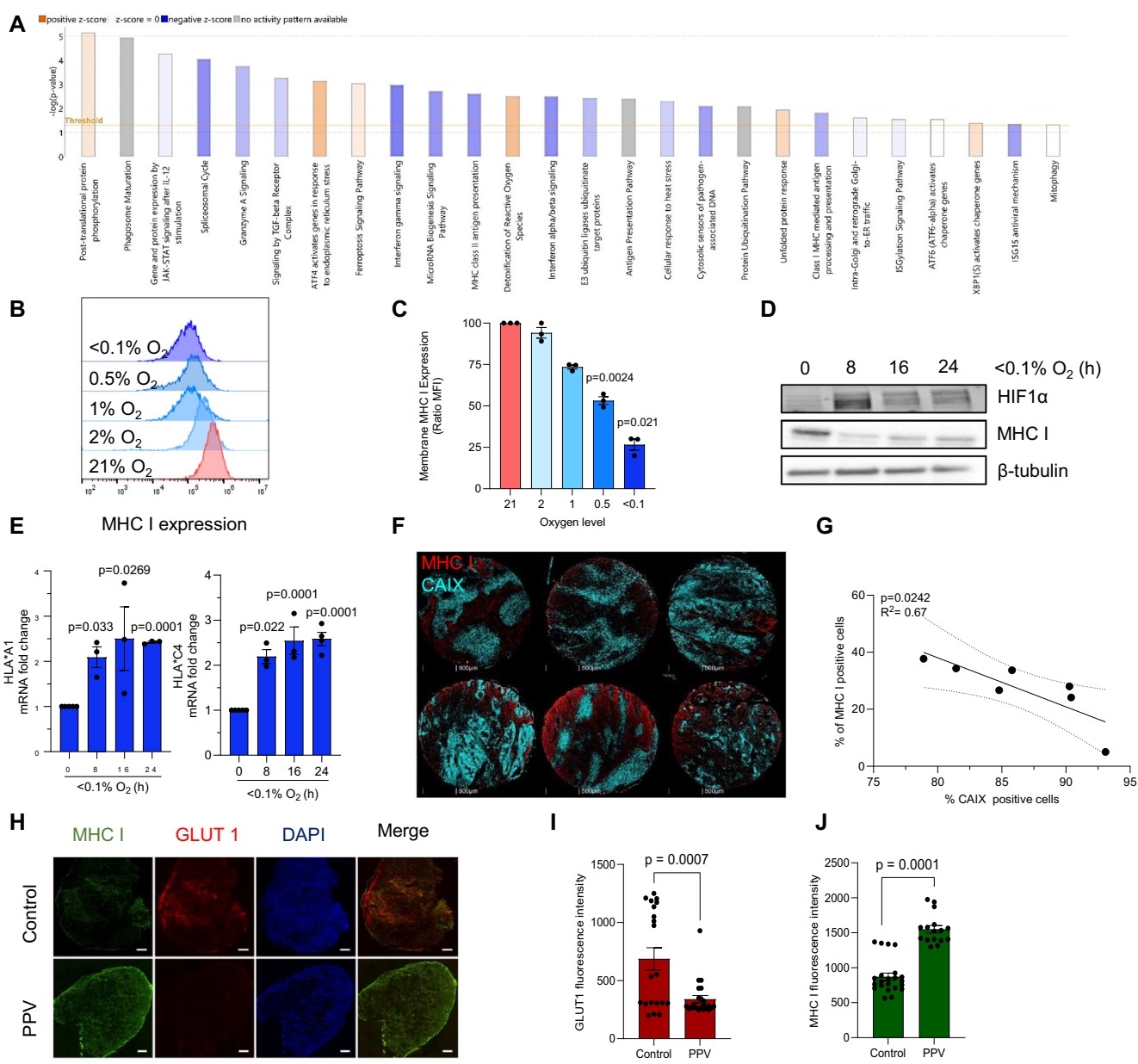

**Figure 2. Hypoxia is associated with reduced levels of MHC I expression.**

(A) Ingenuity Pathway Analysis of differentially expressed proteins in HT29 cells exposed to hypoxia (<0.1% $O_2$) at 24-h time point (−Log2 fold change >1.5). Statistical analysis was determined using Fisher's Exact test ($n = 3$ biological replicates). (B) HT29 cells were exposed to 21, 2, 1, 0.5 or <0.1% $O_2$ for 24 h. MHC I expression on the cell surface was determined using flow cytometry-based analysis. Representative histograms showing MHC I expression at different oxygen levels are shown ($n = 3$ biological replicates). (C) Graph showing mean fluorescence intensity of MHC I expression on the cell surface. Data represent mean ± SEM from three independent biological replicates. Statistical analysis was calculated using paired Student's $t$ test. (D) HT29 cells were exposed to hypoxia (<0.1% $O_2$) for the times indicated. MHC I protein level from whole-cell lysates was detected by western blot ($n = 3$ biological replicates). (E) HT29 cells were exposed to <0.1% $O_2$ for the times indicated. qPCR for *HLA*A1* and *HLA*C4* genes was carried out. 18S served as a housekeeping gene. Data represent mean ± SEM from three biological replicates. Statistical analysis was calculated using unpaired Student's $t$ test. (F) Multiplexed spectral microscopy imaging showing the distribution of CAIX-positive cells (green) and MHC I-positive cells (red) in six different tissue microarrays from patients with colorectal cancer (scale bar: 500 μm). (G) Machine learning-based quantification of CAIX and MHC I-positive cells following multiplex staining, graph showing negative correlation between MHC I-positive cells and CAIX-positive cells. Statistical analysis was determined using simple linear regression test and Pearson correlation. (H) Immunofluorescence staining of a representative tumor tissue of control and papaverine depicting GLUT1 (red), MHC I (green), and DAPI (blue) staining (scale bar: 500 μm). (I) Graph showing the quantification of GLUT1 fluorescence intensity. Data represent mean ± SEM from three different mice and at least two sections were stained for each condition. Statistical analysis was calculated using unpaired Student's $t$ test. (J) Graph showing the quantification MHC I fluorescence intensity. Data represent mean ± SEM from three different mice and at least two sections were stained for each condition. Statistical analysis was calculated using unpaired Student's $t$ test. Source data are available online for this figure.

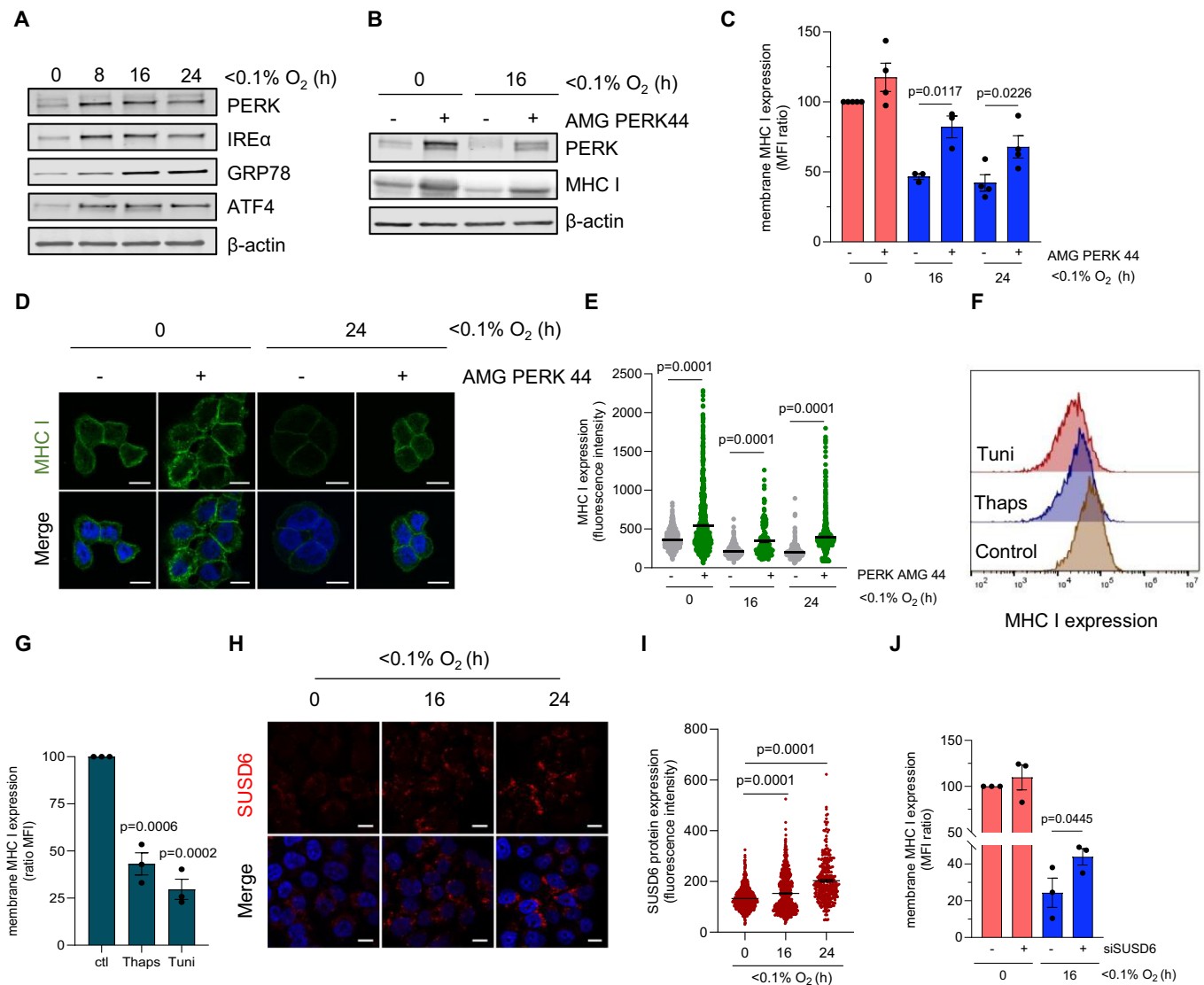

**Figure 3.  Unfolded protein response-dependent MHC I expression under hypoxia.**

(A) HT29 cells were exposed to hypoxia (<0.1% $O_2$) for 8, 16 and 24 h, followed by western blotting for the indicated proteins. A representative western blot of three biological replicates is shown. (B) HT29 cells were treated with PERK inhibitor (10 μM) and exposed to normoxia or hypoxia (<0.1% $O_2$) for 16 h followed by western blotting as indicated. PERK inhibition was confirmed by the absence in the electrophoretic mobility shift of PERK ($n = 3$ biological replicates). (C) HT29 cells were treated with PERK inhibitor (AMG PERK 44, 10 μM) and exposed to normoxia or hypoxia (<0.1% $O_2$) for 16 and 24 h followed by flow cytometry. Graph showing mean fluorescence intensity of MHC I expression on the cell surface. Data represent mean ± SEM from three biological replicates. Statistical significance was determined using unpaired Student's *t* test. (D) HT29 cells were treated with PERK inhibitor (10 μM) and exposed to normoxia or hypoxia (<0.1% $O_2$) for 16 and 24 h. Cells were fixed and stained for MHC I by immunofluorescence assays. Representative images are shown (scale bar: 10 μm) ($n = 3$ biological replicates). (E) Graph showing the quantification of the fluorescence intensity of MHC I expression in HT29 cells. Data represent mean ± SEM from three biological replicates. Statistical significance was determined with unpaired Student's *t* test. (F) HT29 cells were treated with Thapsigargin (Thaps, 2 μM) and Tunicamycin (Tuni, 5 μg/ml) for 24 h. Representative histograms showing MHC I expression determined using flow cytometry ($n = 3$ biological replicates). (G) Graph showing mean fluorescence intensity of MHC I expression on the cell surface. Data represent mean ± SEM from three biological replicates. Statistical significance was determined with unpaired Student's *t* test. (H) HT29 cells were exposed to hypoxia (<0.1% $O_2$) for the times indicated. Immunofluorescence staining for SUSD6 was then carried out. Representative images from three biological replicates are shown (scale bar: 10 μm), SUSD6 (red), DAPI (blue). (I) Graph showing the quantification of the fluorescence intensity of SUSD6 expression in HT29 cells. Data represent mean ± SEM from three biological replicates. Statistical analysis was determined using unpaired Student's *t* test. (J) HT29 cells were transfected with siSUSD6 and exposed to hypoxia (<0.1% $O_2$) for 16 h. Mean fluorescence intensity of MHC I expression on the cell surface determined using flow cytometry is shown. Data represent mean ± SEM from three biological replicates. Statistical analysis was calculated using paired Student's *t* test. Source data are available online for this figure.

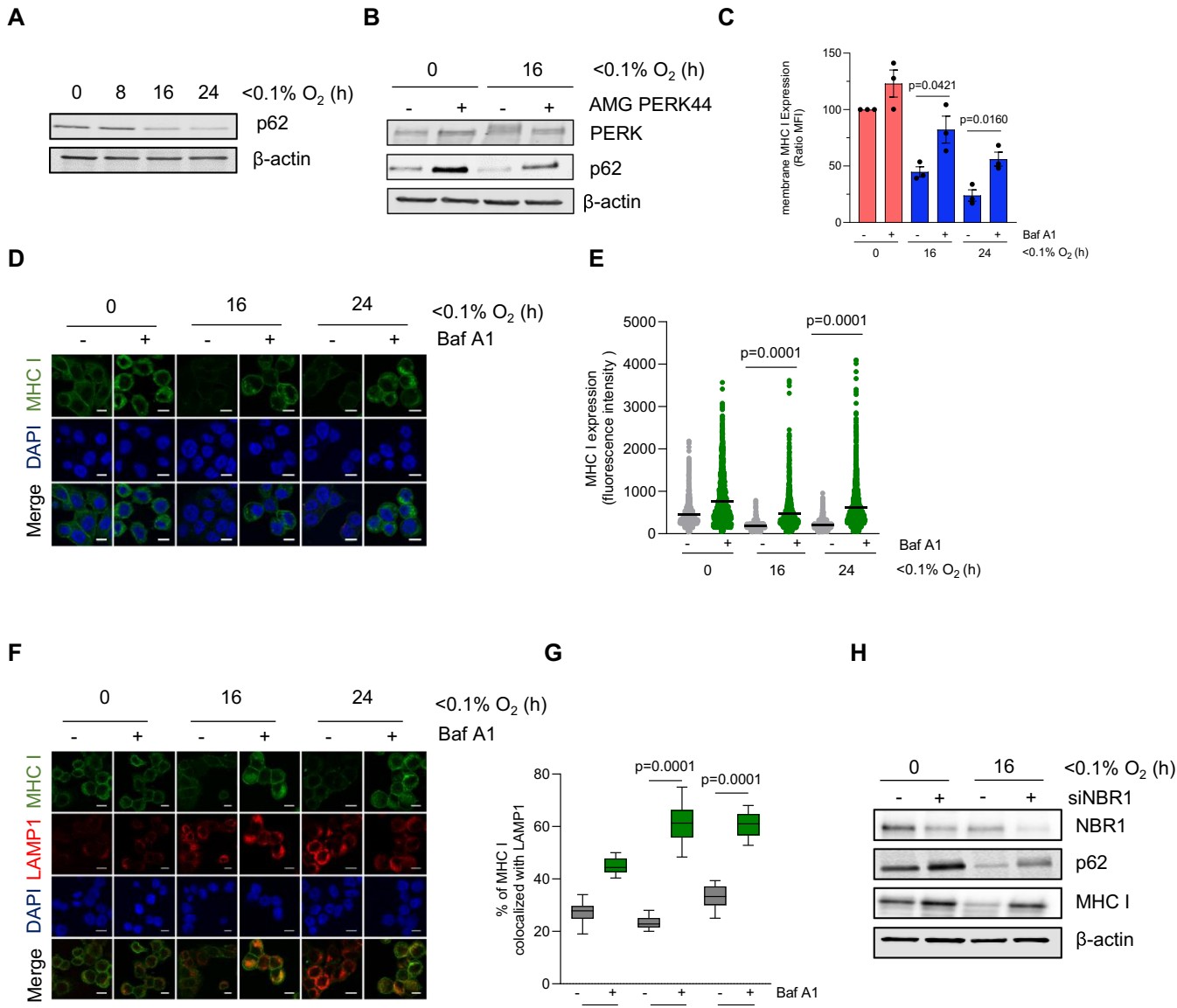

**Figure 4. Activation of autophagy induces MHC I degradation under hypoxia.**

(A) HT29 cells were exposed to hypoxia (<0.1% $O_2$) for 8, 16, and 24 h, western blotting was then carried out with antibodies indicated. A representative western blot of three biological replicates is shown. (B) HT29 cells were treated with PERK inhibitor (10 µM) and exposed to normoxia or hypoxia (<0.1% $O_2$) for 16 h followed by western blotting as indicated (n = 3 biological replicates). (C) HT29 cells were treated with bafilomycin A1 (50 nM) and exposed to hypoxia (<0.1% $O_2$) for 16 and 24 h. MHC I expression on the cell surface was determined using flow cytometry. Graph showing mean fluorescence intensity of MHC I expression on the cell surface. Data represent mean ± SEM from three biological replicates. Statistical analysis was calculated using unpaired Student's *t* test. (D) HT29 were treated as in part (B), staining for the localization of MHC I was then carried out. Representative images are shown (scale bar: 10 µm). MHC I staining in green, DAPI (blue) (n = 3 biological replicates). (E) Graph showing the quantification of the fluorescence intensity of MHC I expression in HT29 cells. Data represent mean ± SEM from three biological replicates. Statistical analysis was determined using unpaired Student's *t* test. (F) HT29 cells were treated as in (B), staining for MHC I and LAMP1 was then carried out. representative images showing the localization of MHC I (green) relative to LAMP1 positive (red) lysosomes (scale bar: 10 µm) (n = 3 biological replicates). (G) Graph showing the quantification of the percentage of colocalization of MHC I with LAMP1 in HT29 cells. At least 100 cells were quantified in each group. Data are presented as box plot showing the distribution of colocalization percentages, with the median shown as the central line in each box, the interquartile range (IQR) spanning the box, and whiskers extending to the minimum and maximum values within 1.5 times the IQR from three biological replicates. Statistical analysis was determined using unpaired Student's *t* test. (H) HT29 cells were transfected with siNBR1 and exposed to hypoxia (<0.1% $O_2$) for 16 h and subjected to western blotting with antibodies indicated. A representative western blot of three biological replicates is shown. Source data are available online for this figure.

(Kirkin et al, 2009). NBR1 is a cargo receptor that regulates MHC I degradation by autophagy (Yamamoto et al, 2020). To investigate the role for NBR1 in trafficking MHC I to the lysosomes under hypoxia, HT29 cells were transfected with siNBR1 and exposed to normoxia or hypoxia. Interestingly, NBR1 knockdown significantly increased total and plasma membrane levels of MHC I in HT29 cells (Fig. 4H; Appendix Fig. S5H). Collectively, these data demonstrate that MHC I is decreased under hypoxia by an autophagy–lysosomal-dependent pathway.

## Downregulation of MHC I under hypoxia correlates with a decrease in peptide presentation

To determine if the changes in MHC I expression observed under hypoxia resulted in both a decrease as well as a change in antigen presentation, we used LC-MS technology to analyze the immunopeptidome. HCT116 cells were treated either in normoxia (21% O$_2$) or hypoxia for 24 h; cells were then collected and a quantitative analysis of the immunopeptidome was performed (Fig. 5A). Upon acquiring the LC-MS data, peptides were filtered based on the length (8–14 amino acids) and a binding score of less than 2 according to their rank using the NetMHCpan 4.1 algorithm. Overall, cellular exposure to hypoxia resulted in a significant decrease in the total number of peptides binding to MHC I compared to cells exposed only to normoxia (Fig. 5B). Upon analysis of the unique peptides presented in each condition, where a peptide is considered unique when it appears in a single biological replicate, 2521 unique peptides were identified in normoxia versus only 184 unique peptides under hypoxic conditions (Fig. 5C). Among the small subset of peptides unique to hypoxia, an overexpression analysis of their source proteins did not reveal any significant pathways indicating that changes in the global immunopeptidome are likely linked to the upstream changes in MHC I (Appendix Fig. S6A). The length distribution of the peptides, with 9-mers being most prevalent bound to human MHC I, was unchanged between normoxic and hypoxic conditions, indicating that there were no obvious length-related effects (Fig. 5D). A comparison of the sequence logos of peptides binding to each allele showed no global changes in the binding anchor residues between normoxia and hypoxia (Appendix Fig. S6B). The changes observed under hypoxia in the immunopeptidome were consistent across all presenting alleles suggesting no allele-dependent effect (Fig. 5E). Due to this dramatic decrease in unique peptides under hypoxia, we assessed the expression of the other components of the antigen presentation complex after exposure of HT29 cells to hypoxia. The immunoproteasome subunits, LMP2, LMP7, and the transporter-associated protein TAP2 were not induced by hypoxic conditions but were induced by INFγ treatment (Fig. 5F; Appendix Fig. S6C). Multiplexed spectral fluorescence microscopy in colorectal cancer patient tissue microarray confirmed the absence of induced expression of LMP2 and TAP2-positive cells in hypoxic areas stained with the hypoxia marker CAIX (Fig. 5G,H). Taken together, these data reveal that the decrease of MHC I under hypoxia is coupled with an absence of expression of other components of the antigen presentation complex and results in a decrease in the number of presented antigens in the tumor.

## Blocking autophagy under hypoxia induces a change in the immunopeptidome and reveals new antigens

We then explored whether the changes in MHC I expression observed after the inhibition of autophagy in hypoxic conditions resulted in the generation of antigens as well as the restoration of antigens found under normoxic conditions. HT29 cells were treated either in normoxia alone, normoxia and Baf A1, hypoxia alone or hypoxia and Baf A1 for 24 h. Immunopeptidomics and label-free quantitative proteomics were carried out in parallel (Fig. 6A). As expected, at the peptide level, we observed a significant decrease in the total number of peptides when cells were exposed to hypoxia compared to normoxia. However, under both normoxic and hypoxic conditions, Baf A1 treatment significantly increased the number of peptides presented by MHC I (Fig. 6B). The length distribution analysis again showed no length-altering effects or alternative peptide cleavage, displaying a typical human MHC I length distribution (Fig. 6C). This was also reflected in the MHC I binding analysis (Fig. 6D; Appendix Fig. S7). Analysis of unique peptides indicated that the changes in antigen presentation in each condition were present irrespective of length or allele. (Fig. 6E). One unique peptide appeared in hypoxia while 46 unique peptides were present when cells were treated with Baf A1 under hypoxia. Importantly, these peptides were not present in the normoxia + Baf A1 condition indicating their specificity to hypoxia. In addition, 716 of the peptides present in normoxia were rescued upon treatment with Baf A1 under hypoxia. Together, our results suggest that the inhibition of autophagy could reverse hypoxia-mediated immune evasion through the presentation of antigens (Fig. 6E). MHC I peptides originate from the cleavage of endogenous source proteins and the expression of these proteins may modulate the number of peptides that originate from their respective source protein. Therefore, we evaluated if changes in the proteome were translated into changes in antigen presentation. A differential expression analysis of the proteomics data under different treatment conditions was carried out (Appendix Fig. S8A,B). Peptides in the immunopeptidomics data which were considered quantifiable using Progenesis software analysis (Waters Corporation) were summed according to their source proteins. The Venn diagram is representative of the overlap between the source proteins from the proteomics and the immunopeptidomics data based on their successful alignment parameters in Progenesis. 762 proteins which overlapped between the proteome and the immunopeptidome were used to represent the overall changes occurring in the omics data (Fig. 6F). By plotting the ratio between these parameters, we found that changes in the proteome minimally influenced the changes observed in the immunopeptidome and that both the down-regulation and upregulation of proteins can result in the presentation of MHC I peptides and vice versa (Fig. 6G; Appendix Fig. S9). The analysis of the proteome and immunopeptidome under hypoxia compared to normoxia revealed a significant downregulation in the immunopeptidome compared to the proteome (bottom left quadrant) (Fig. 6G). Interestingly, when cells were treated with Baf A1 under hypoxia, an upregulation in the immunopeptidome and the proteome resulted (Fig. 6H). In support of the lack of impact on protein changes influencing antigen presentation, we found that many proteins that were unchanged in the proteome still showed relevant changes in the immunopeptidome. In conclusion, this analysis demonstrates that inhibition of autophagy can rescue MHC I peptides in hypoxia and that the changes in antigen presentation are largely due to the altered regulation of MHC I and not changes in expression of the source proteins.

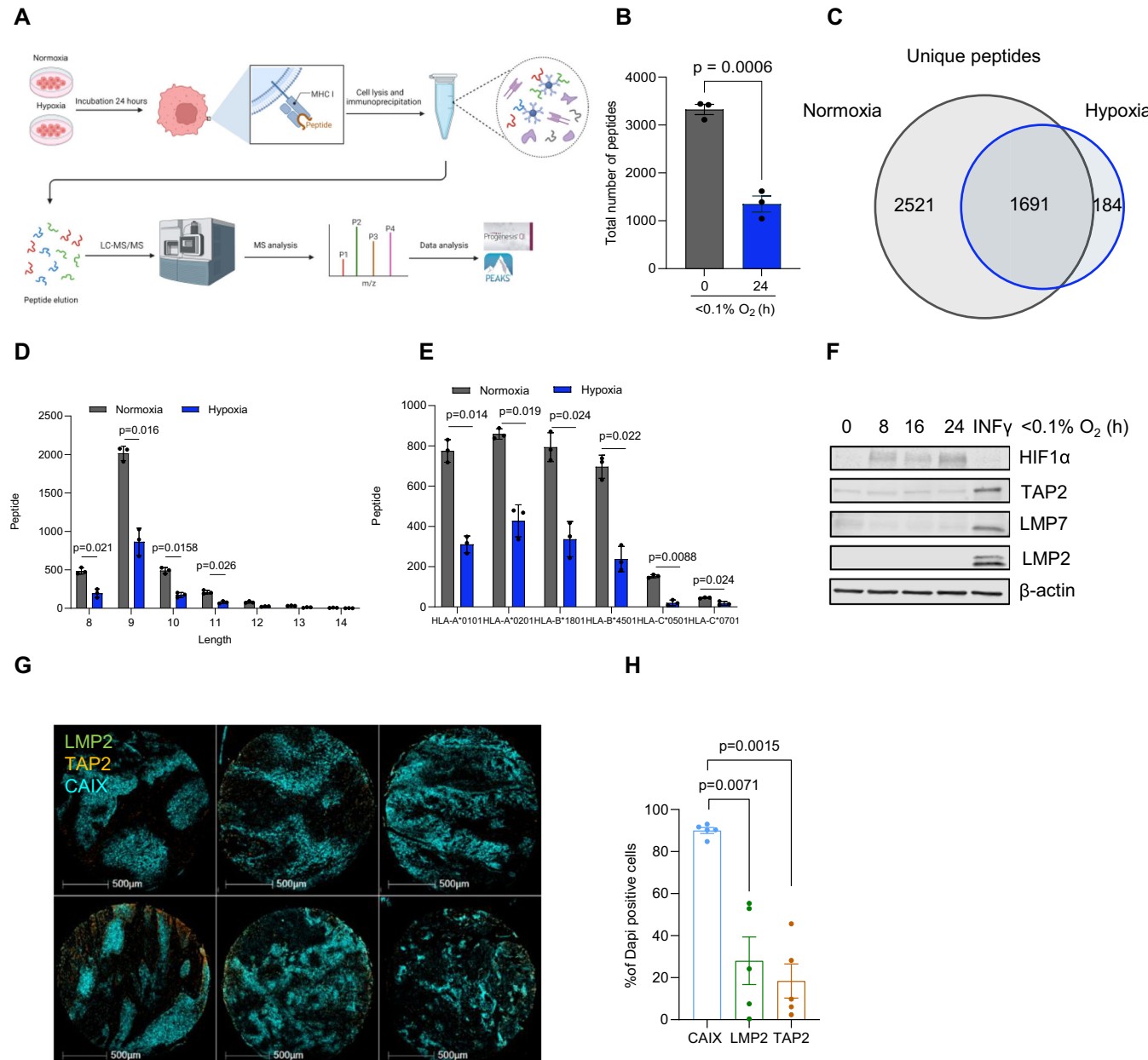

**Figure 5. Downregulation of MHC I under hypoxia is correlated with a decrease in peptide presentation.**

(A) Schematic overview of the immunopeptidomics experiment workflow for HCT116 cells exposed to normoxia (21% $O_2$) or hypoxia (<0.1% $O_2$) for 24 h. This scheme was created with Biorender.com. (B) The total number of MHC I peptides of HCT116 cells exposed to normoxia (21% $O_2$) or hypoxia (<0.1% $O_2$) for 24 h. Data represent mean ± SEM from three biological replicates. Statistical significance was determined using unpaired Student's $t$ test. (C) Unique MHC I peptides in normoxia (21% $O_2$) versus hypoxia (<0.1% $O_2$) at 24-h time point. (D) Length frequency distribution of MHC I peptides in HCT116 cells exposed to normoxia (21% $O_2$) or hypoxia (< 0.1% $O_2$). Data represent mean ± SEM from three biological replicates. Statistical significance was determined with paired Student's $t$ test. (E) Allele-binding distribution in normoxia (21% $O_2$) and hypoxia (< 0.1% $O_2$) at 24-h time point. Data represent mean ± SEM from three biological replicates. Statistical significance was calculated with paired Student's $t$ test for each indicated allele. (F) HT29 cells were exposed to <0.1% $O_2$ for the times indicated and western blotting carried out with the antibodies indicated. INFγ (100 ng/ml) was used as a positive control ($n = 3$ biological replicates). (G) Multiplexed spectral microscopy imaging showing the distribution of CAIX-positive cells (blue), TAP2 (orange) and LMP2 (green) in six different tissue microarrays from patients with colorectal cancer. The staining was performed at the same time for CAIX, MHC I, TAP2 and LMP2 and the analysis was conducted separately and presented as indicated in Figs. 2F and 5G (scale bar: 500 μm). (H) Machine learning-based quantification of CAIX high, LMP2, TAP2-positive cells following multiplex staining in tissue microarrays from patients with colorectal cancer. Data represent mean ± SEM from five different tissue microarrays. Statistical analysis was calculated using paired Student's $t$ test. Source data are available online for this figure.

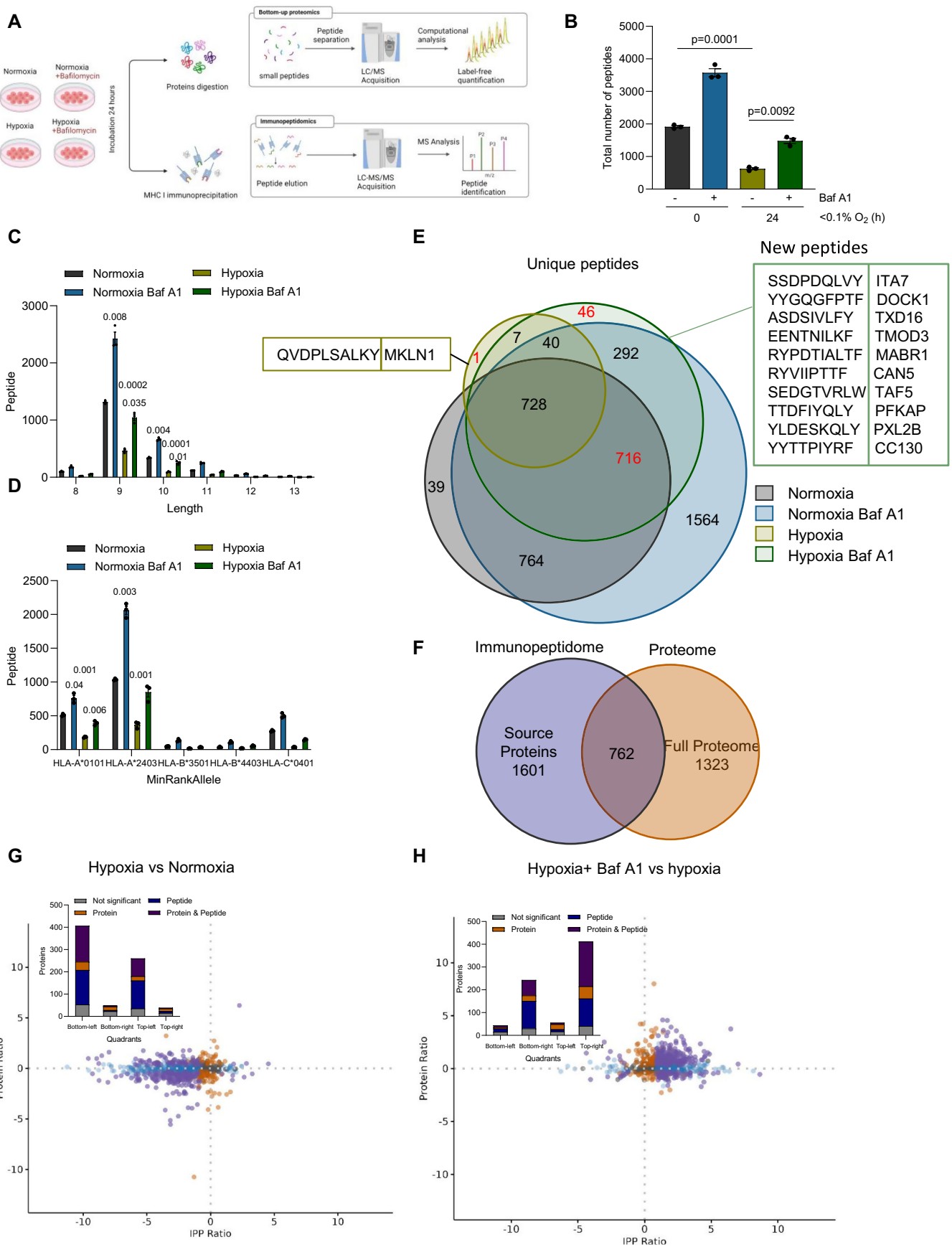

**Figure 6. Blocking autophagy under hypoxia induces a change in the immunopeptidome repertoire and reveals new antigens.**

(A) Schematic overview of the proteomic and immunopeptidomics experiments workflow for HT29 cells exposed to normoxia or hypoxia (< 0.1% $O_2$) and treated with bafilomycin A1 (50 nM) for 24 h. This scheme was created with Biorender.com. (B) Graph showing the total number of MHC I peptides of HT29 cells for the indicated conditions. Data represent mean ± SEM from three biological replicates. Statistical analysis was calculated using paired Student's *t* test. (C) Graph showing the length frequency distribution of MHC I peptides in HT29 cells for the indicated conditions. Data represent mean ± SEM from three biological replicates. Statistical significance was determined with paired Student's *t* test. (D) Graph showing the allele-binding distribution in HT29 cells for the indicated conditions. Data represent mean ± SEM from three biological replicates. Statistical significance was calculated with paired Student's *t* test for each indicated allele. (E) Venn diagram exhibiting the unique MHC I peptides for the different indicated conditions in HT29 cells. The sequence of the unique peptide in hypoxia and the sequences of the top binders peptides in the hypoxia and bafilomycin condition based on a threshold below 0.5 rank binding score are presented. (F) Venn diagram showing the overlap of proteome and immunopeptidome source proteins. (G) Correlation plot of the proteome protein and the immunopeptidome peptide ratio for hypoxia (< 0.1% $O_2$) versus normoxia. Significantly differentially expressed proteins and peptides are indicated by their colors. A bar chart to quantify proteins in each quadrant is presented next to each plot. A *P* value significance threshold of <0.05 was used to indicate significantly differentially expressed proteins from the proteome analysis (orange), differentially expressed source proteins from the immunopeptidome analysis (blue), and proteins where both proteomic and immunopeptidomics analyses were significant (purple). Statistical analysis was determined using one-way ANOVA test. (H) Correlation plot of the proteome protein and the immunopeptidome peptide ratio for hypoxia (< 0.1% $O_2$) versus hypoxia (< 0.1% $O_2$) + bafilomycin A1. Significantly differentially expressed proteins and peptides are indicated by their colors. A bar chart to quantify proteins in each quadrant is presented next to each plot. A *P* value significance threshold of <0.05 was used to indicate significantly differentially expressed proteins from the proteome analysis (orange), differentially expressed source proteins from the immunopeptidome analysis (blue), and proteins where both proteomic and immunopeptidomics analyses were significant (purple). Statistical analysis was determined using one-way ANOVA test. Source data are available online for this figure.

## Modulation of hypoxia changes the immunopeptidome in vivo and enhances T-cell-mediated cytotoxicity ex vivo

We finally sought to determine whether the changes of MHC I expression in vivo also correlate with changes in the immunopeptidome following treatment with the mitochondrial complex-I inhibitor papaverine. Using immunopeptidomics, we isolated pure MHC I-bound peptides from murine CT26 tumors that were untreated or treated with 2 mg/kg papaverine (Fig. 7A; Appendix Fig. S10A). Analysis of the LC-MS data from these two groups of tumors demonstrated a significant increase in the total number of peptides in tumors treated with papaverine compared to untreated control tumors (Fig. 7B). The summed intensity of all presented peptides also showed a substantial increase in the papaverine-treated tumors, indicating not only that there are more peptides being presented but that they are also more abundant (Fig. 7C). The distribution of peptides binding to each allele showed that most peptides bind significantly to H2-Dd and H2-Kd alleles (Fig. 7D). Comparison of the unique peptides derived from the untreated or papaverine-treated tumors uncovered 808 unique peptides in the papaverine condition compared to 130 unique peptides in the control (Fig. 7E). These peptides originated from 429 unique source proteins that were quantified in the papaverine condition (Appendix Fig. S10B). Among the subset of peptides unique to papaverine, overexpression analysis of source proteins revealed several pathways associated with transcription indicating that changes in the global immunopeptidome are likely linked to both changes in transcription of their source proteins induced by papaverine in the switch from a hypoxic to oxic state as well as changes in MHC I expression (Appendix Fig. S10C). This result is distinct from the results found when treating cells under oxic or hypoxic conditions with autophagy inhibitors where changes in protein expression did not result in substantial changes in the immunopeptidome. These data demonstrate that decreasing mitochondrial metabolism in tumor cells with papaverine treatment reduces the hypoxic state of tumor cells and induces a change in the immunopeptidome, uncovering new targetable antigens. Finally, given that MHC I-Antigen presentation plays a critical role in regulating CD8[+]T-cell activation and function (Peaper and Cresswell, 2008; Dockree et al, 2017), we determined whether

inhibiting MHC I degradation under hypoxic conditions by blocking autophagy or the PERK arm of the UPR could enhance antigen-specific CD8[+]T-cell responses. We conducted co-culture experiments with CD8[+]OT-I T cells and MC38 colorectal tumor cells expressing the Ova peptide (Fig. 7F). These data demonstrate that MC38-Ova cells pretreated with Baf A1 or PERK inhibitor AMG44 under hypoxia (<0.1% $O_2$) were more vulnerable to OT-1 T-cell-mediated killing compared to MC38-Ova cells co-cultured with OT-1 T cells alone (Figs. 7G and 8). Together, these results support the conclusion that preventing MHC I degradation under hypoxia enhances immune recognition of hypoxic cells and improves the effectiveness of T-cell killing.

## Discussion

In this study, we have shown that tumor hypoxia activates an immune escape mechanism through the downregulation of the MHC I expression and a subsequent decrease in antigen presentation that correlate with an exclusion of CD8[+]T cells. We propose that hypoxic conditions induce the UPR and that the PERK arm of the UPR signals for the induction of SUSD6, leading to the autophagic targeting of MHC I by the adaptor protein NBR1, that result in MHC I downregulation and antigen presentation. The inhibition of PERK with a specific small molecule inhibitor or the direct inhibition of autophagy under hypoxic conditions rescued MHC I expression and restored the immunopeptidome as well as generated new potential antigens (Fig. 8). Previous isolated studies have linked hypoxia to immune evasion through the inhibition of INF-γ dependent genes, which are essential for recruitment of effector immune cells such as CD8[+]T cells into tumors or potential decreases at the transcriptional level of MHC I expression in a HIF dependent manner in malignant cells under hypoxic (10% $O_2$) conditions (Murthy et al, 2019; Sethumadhavan et al, 2017). In our study, we identified a link between the PERK arm of the UPR in hypoxia inducing the activation of autophagy and the degradation of MHC I. Prior studies have reported that ER stress induced by tunicamycin, palmitate or glucose deprivation decreases MHC I surface expression (Granados et al, 2009), and reduced surface MHC I levels have also been detected in human cells overexpressing transcriptionally active isoforms of

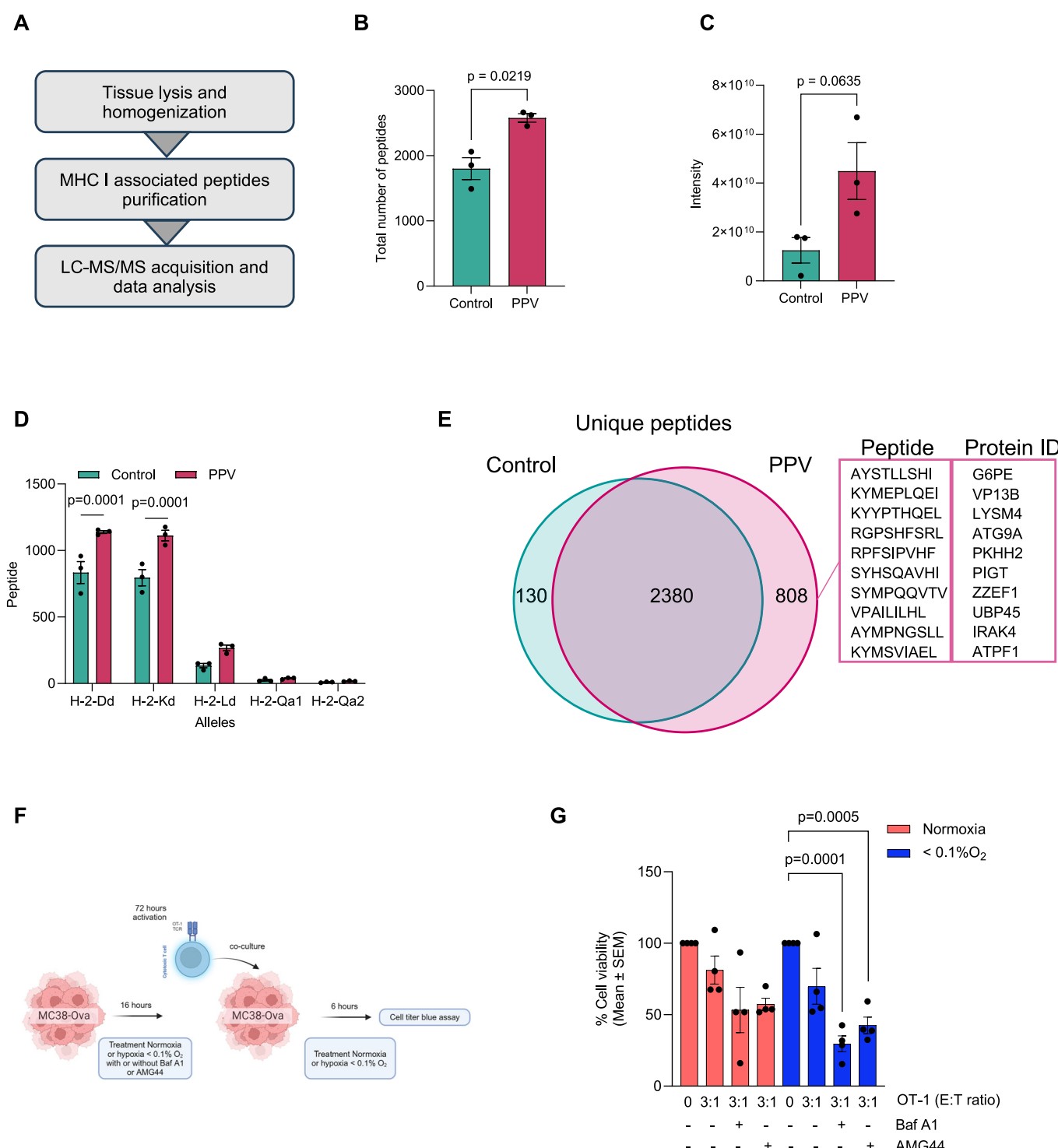

**Figure 7. Modulation of hypoxia changes the immunopeptidome in vivo and enhances T-cell-mediated cytotoxicity ex vivo.**

(A) Overview of the immunopeptidomics experiment on tumor tissues. (B) Graph showing the total number of MHC I-bound peptides for the indicated conditions. Data represent mean ± SEM from three different mice. Statistical analysis was determined using paired Student's *t* test. (C) Graph showing MHC I-bound peptides intensity for the indicated conditions. Data represent mean ± SEM from three different mice. Statistical analysis was determined using unpaired Student's *t* test. (D) Graph showing the allele-binding distribution of MHC I peptides in control and papaverine-treated tumors. Data represent mean ± SEM from three different mice. Statistical analysis was determined using paired Student's *t* test. (E) Venn diagram showing unique MHC I-bound peptides in control versus papaverine. The top 10 strong binders peptides in papaverine condition based on a threshold below 0.5 rank binding score are presented. (F) Schematic overview of the in vitro cytotoxicity assay workflow using OT-1 CD8+T cells and MC38-Ova tumor cells. (G) Graph showing the percentage of cell viability of MC38-Ova normalized to the negative control without OT-1 T cells for the indicated conditions. Data represent mean ± SEM from four biological replicates. Statistical analysis was calculated using a two-way ANOVA test. Source data are available online for this figure.

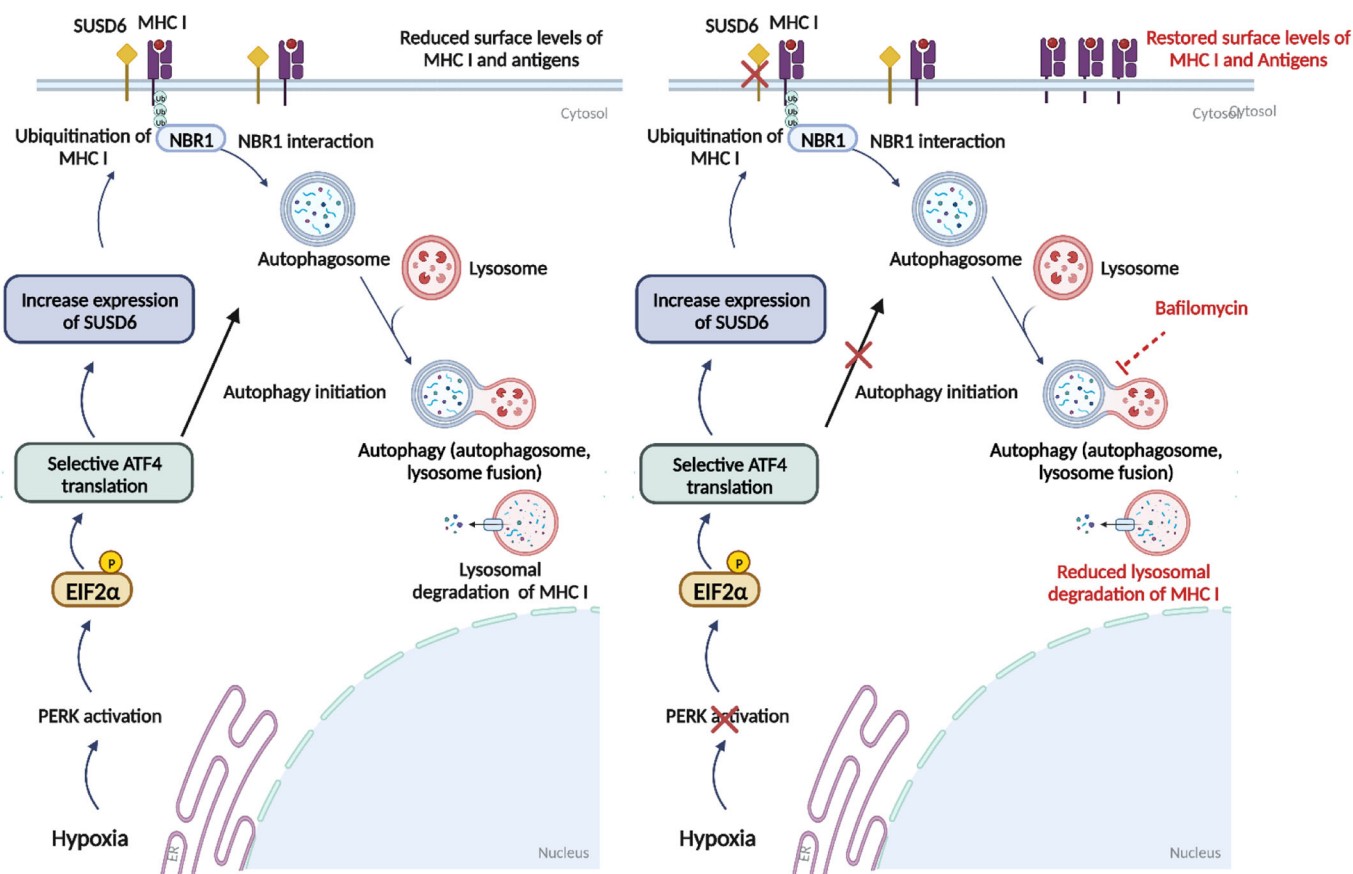

**Figure 8.** Graphical summary showing the detailed mechanism of MHC I downregulation under hypoxia.

UPR-activated transcription factors ATF6 and XBP1 (De Almeida et al, 2005; de Almeida et al, 2007). Since the UPR provokes the degradation of ER-localized mRNA (Hollien and Weissman, 2006) accelerated decay of MHC I mRNA might have been responsible for the reduction of MHC I expression. However, the presence of high levels of MHC I transcripts under hypoxia excludes this possibility. Furthermore, our results reveal that autophagy is a key regulator of MHC I expression under oxic and particularly hypoxic conditions, and that autophagy activation induces lysosomal degradation through the cargo receptor NBR1. Inhibition of autophagy either pharmacologically or genetically restored MHC I expression under hypoxia. Several recent studies support the importance of autophagy in regulating MHC I, including the inhibition of basal autophagy in pancreatic ductal adenocarcinoma cells which restores MHC I cell surface expression and increases tumor infiltration of CD8[+]T cells in mice (Yamamoto et al, 2020; Münz, 2021; Kong et al, 2023). Our findings suggest a functional role of the membrane protein SUSD6 in inhibiting MHC I expression under hypoxic conditions. We have found that the expression of SUSD6 is induced under hypoxia mainly through the UPR. In addition, knocking down SUSD6 rescued MHC I expression under hypoxia. In a recent study, SUSD6 and TMEM127 have been identified as negative regulators for MHC I expression through their recruitment of E3 ubiquitin ligase WWP2, resulting in lysosomal degradation of MHC I (Chen et al, 2023). Under hypoxic conditions, SUSD6, in a TEM127-independent manner, serves as a regulator at the membrane level to recruit certain E3 ligases including

STUB1 to MHC I for its ubiquitination, resulting in recognition and binding by NBR1 and autophagic degradation. Further studies should explore in more detail how STUB1, or other E3 ligases, lead to the ubiquitination of MHC I under hypoxia which seems to occur independently of TEM127 in MHC I degradation.

We have also demonstrated how the degradation of MHC I under hypoxic conditions may influence T-cell surveillance by altering the immunopeptidome. The changes observed at the peptide level through the inhibition of autophagy, or the alleviation of hypoxia led to the presentation of new potential antigens. The inhibition of autophagy under hypoxia rescued peptides which are typically presented under normoxic conditions and more importantly uncovered new unique MHC I peptides specific to hypoxia. In addition, the changes observed in the immunopeptidome in vivo are likely influenced by both changes in the proteome and the MHC I expression, that could subsequently improve CD8[+]T cells infiltration and T-cell-mediated cytotoxicity. These results emphasize a novel therapeutic strategy for immunologically "cold tumors" to enhance the efficacy of cancer immunotherapy. Further analysis using proteogenomics approaches are planned to characterize the presence of tumor-specific neoantigens and whether their expression mimics the changes we have observed. Immunopeptidomics has permitted a better and more accurate understanding of the complex interplay between the true tumor antigenic landscape and the immune response (Kraemer et al, 2023; Olsson et al, 2021). Furthermore, it has enabled us to better understand the dynamic

nature of antigen presentation under different treatment conditions (Tailor et al, 2022; Murphy et al, 2019; Shraibman et al, 2016).

The ability to counteract immune evasion is a critical component of this work and will require some further study to fully define how the inhibition of autophagy or reducing hypoxia performs in a therapeutic setting. However, it is clear that the field of cancer immunotherapy is shifting its focus from relatively nonspecific regulators of the immune system like checkpoint inhibition to therapies which target specific p-MHC complexes like TCR T-cell therapies and cancer vaccines. Therefore, while tumor-specific neoantigens may have the potential to be presented as observed in the immunopeptidome under "normal" conditions, factors like hypoxia greatly influence the global tumor antigenic landscape. As such, capturing these changes and mitigating them may determine the success or failure of immune checkpoint therapies.

# Methods

## Tools and reagents

A list of reagents (antibodies, drugs, siRNAs, primers and chemicals) is provided in Reagents and Tools.

## Methods and protocols

### Cell culture

HT29, HCT116, Calu6, A549, RKO, and RKO$^{HIF-/-}$ Cells were grown in DMEM supplemented with 10% FBS and cultured in humidified incubators at 37 °C and 5% $CO_2$. CT26 and MC38-Ova cells were grown in RPMI supplemented with 10% FBS at 37 °C and

**Reagents and tools table**

| Reagent/resource | Reference or source | Identifier or catalog number |
| --- | --- | --- |
| **Experimental models** | | |
| HT29 | ECACC | 91072201 |
| HCT116 | ATCC | CCL-247 |
| Calu6 | ATCC | HTB-56 |
| A549 | ATCC | CRM-CCL-185 |
| CT26 | ATCC | CRL-2638 |
| MC38-Ova | Prof. Nicholas Denko lab | |
| RKO | Dang et al, 2006 | |
| RKO $^{HIF-/-}$ | Dang et al, 2006 | |
| Balb/c (*M. musculus*) | Charles River | |
| C57BL/6 (*M. musculus*) | Charles River | |
| **Antibodies** | | |
| GRP78 | BD Biosciences | 610978 |
| HIF1α | BD Biosciences | 610959 |
| HLA A, B, C (MHC I) | Abcam | ab70328 |
| LMP2 | Abcam | ab3328 |
| β-tubulin | Abcam | ab6046 |
| p62 | Cell Signaling | 88588 |
| LC3B | Cell Signaling | 3868 |
| ATG5 | Cell Signaling | 2630 |
| ATG12 | Cell Signaling | 4180 |
| ATF4 | Cell Signaling | 11815 |
| IREα | Cell Signaling | 3294 |
| PERK | Cell Signaling | 3192 |
| LMP7 | Cell Signaling | 13635 |
| NBR1 | Santa Cruz | sc-130380 |
| β-actin | Santa Cruz | sc-69879 |
| TAP2 | Invitrogen | PA5-37414 |
| SUSD6 | Invitrogen | PA5-56481 |
| LAMP1 | Cell Signaling | 9091 |
| GLUT1 | Abcam | ab115730 |

| Reagent/resource | Reference or source | Identifier or catalog number |
|---|---|---|
| CD8 | Cell Signaling | 64786 |
| Goat anti-mouse IgG (H + L) | Licor Biosciences | IRDye800RD |
| Goat anti-rabbit IgG (H + L) | Licor Biosciences | IRDye680CW |
| Anti-rabbit IgG HRP-linked | Merck | 31460 |
| Anti-mouse IgG, HRP-linked | Merck | 31430 |
| HLA A, B, C | Biolegend | 311413 clone w6/32 |
| MHC I antibody clone 34.1.2 s | ATCC | Hybridoma cells HB-79 |
| MHC I antibody clone w6/32 | ATCC | Hybridoma cells HB-95 Immunopeptidomics experiment |
| CAIX | Abcam | ab15086 |
| LMP2 | ProteinTech | 14544-1-AP |
| CD8 | Dako | M7103 |
| E-cadherin | Cell Signaling | 3195 |
| CD3 | Invitrogen | MA5-51796 |
| CD28 | Invitrogen | 740009 M |
| Goat anti-rabbit Secondary Antibody, Alexa Fluor 568 | Invitrogen | A11036 |
| Goat anti-mouse Secondary Antibody, Alexa Fluor 488 | Invitrogen | A28175 |
| Goat anti-Rabbit Secondary Antibody, Alexa Fluor 647 | Invitrogen | A21244 |
| **siRNAs (gene target)** | Dharmacon | |
| ATG5 | L-004374-00 | ON-TARGETplus Human ATG5 siRNA - SMARTpool |
| ATG12 | L-010212-00 | ON-TARGETplus Human ATG12 siRNA - SMARTpool |
| NBR1 | L-010522-00 | ON-TARGETplus Human NBR1 siRNA - SMARTpool |
| SUSD6 | L-020259-01 | ON-TARGETplus Human SUSD6 siRNA - SMARTpool |
| TMEM127 | L-01021-01 | ON-TARGETplus Human TMEM127 siRNA - SMARTpool |
| STUB1 | L-007201-00 | ON-TARGETplus Human STUB1 siRNA - SMARTpool |
| **RT-qPCR primers** | **Forward** | **Reverse** |
| **18S** | TAGAGGGACAAGTGGCGTTC | CGGACATCTAAGGGCATCAC |
| **HLA-A1** | ACTCAGATTCTCCCCAGACG | CCTCGCTCTGGTTGTAGTAGC |
| **HLA-A2** | TGAAGGCCCASTCACAGAY | GBGTGGTGRGTCATATGYGT |
| **HLA-C4** | TCTTCCCAGCCYACCATC | ACAGGTCAGTGTGGGGACA |
| **SUSD6** | AGTGGAAACCAGCCATGGAG | CCACTATAGACAGCGTGGGG |
| **TMEM127** | CTGGTTGCACATCCACGGA | CTGGGGATTCATGCAGAAATCT |
| **Chemicals, enzymes, and other reagents** | | |
| Bafilomycin A1 | Merck | B1793 |
| Chloroquine | Sigma-Aldrich | C6628 |
| Thapsigargin | MP Biomedicals | 02158999-CF |
| Tunicamycin | MP Biomedicals | 02150028-CF |
| AMG PERK 44 | Tocris | 5517 |
| Ceapin-A7 | Sigma-Aldrich | SML2330 |
| 4μ8c | Sigma-Aldrich | 412512 |
| Prolong® Gold with Dapi | Invitrogen | P36962 |
| Hoechst | Invitrogen | |
| Lipofectamine RNAiMax | Thermo Fisher Scientific | 13778075 |
| Goat serum | Merck | G9023 |
| CD8 microbeads | Miltenyi | 130-104-075 |
| 2-Mercaptoethanol | Gibco | 21985023 |

| Reagent/resource | Reference or source | Identifier or catalog number |
|---|---|---|
| Recombinant mouse IL-2 | Miltenyi | 130-120-662 |
| SIINFEKL peptide | Genscript | SC1208 |
| Celltiter-Blue | Promega | G8080 |
| Clarity western ECL Substrate | Bio-Rad | 1705061 |
| Prolong® Gold antifade mountant | Thermo Fisher Scientific | P36934 |
| TRizol | Merck | T9424 |
| SYBR green | Thermo Fisher Scientific | 4364344 |
| Verso cDNA synthesis kit | Thermo Fisher Scientific | 11832113 |
| VECTASHIELD® Vibrance™ Antifade Mounting Medium | Vector Laboratories | H-1700-10 |
| **Software** | | |
| Progenesis QI for proteomics | Waters | |
| Ingenuity Pathway Analysis | Quiagen | |
| PEAKS | Bioinformatics solutions | |
| FlowJo v10.8.1 | BD Biosciences | |
| Halo® | Indica labs | |
| Fiji | ImageJ Wiki | |
| GraphPadPrism10 | GraphPad | |
| R studio v4.2.0 | Posit | |
| Image Studio | Licor | |

5% $CO_2$. Cell lines were regularly tested and verified to be mycoplasma negative using myco-strip (Invivogen).

### Hypoxia treatment

Bactron II and BactronEZ anaerobic chambers (Shel Lab) were used for <0.1% $O_2$. An M35 hypoxia workstation (Don Whitley Ltd) was used for 0.5%, 1%, and 2% $O_2$. All experiments at <0.1% $O_2$ were plated in glass dishes. All hypoxic treatments were harvested within the chamber using equilibrated buffers. Oxygen levels were confirmed with the Oxylite probe (Oxford Optronix).

### Immunoblotting

Cells were collected and lysed in UTB (9 M urea, 75 mM Tris–HCl pH 7.5, 0.1% 5 M β-mercaptoethanol) and briefly sonicated. Primary antibodies were GRP78, HIF1α, MHC I, LMP2, β-tubulin, p62, LC3B, ATG5, ATG12, ATF4, IREα, PERK, LMP7, NBR1, β-actin, TAP2, SUSD6. Secondary antibodies were IRDye800RD Goat anti-Mouse IgG (H + L) and IRDye680CW Goat anti-Rabbit IgG (H + L), anti-rabbit IgG, horseradish peroxidase (HRP)-linked, anti-mouse IgG, HRP-linked. Images were obtained by chemiluminescence using a ChemiDoc (Bio-Rad) or Odyssey IR imaging technology (LI-COR Biosciences).

### Quantitative reverse-transcription PCR

Total RNA was extracted from cells using TRIzol and cDNA was prepared using the Verso cDNA synthesis kit. Quantitative PCR was performed with SYBR Green PCR master mix. The quantity of mRNA was determined using ΔΔCT method and normalized by 18S used as reference gene.

### Immunofluorescence staining

Cells grown on coverslips were fixed with 4% paraformaldehyde and permeabilized with 0.1% triton X-100. Samples were then blocked with 1% BSA in 0.1% Tween-20 in PBS before incubation with primary antibodies overnight at 4 °C. After washing three times with PBS, cells were incubated in secondary antibodies for 1 h at room temperature. Coverslips were mounted using mounting medium Prolong® Gold with DAPI and imaged on a LSM710 confocal microscope (Carl Zeiss) using a 63× objective. Image processing and quantification were performed using ImageJ.

### Flow cytometry

For surface MHC I staining of human cell lines, cells were stained with Alexa fluor 488 anti-human HLA-A, B, C antibody at 1:100 dilution for 30 min on ice in the dark and washed with PBS plus 1% FBS and 2% EDTA 0.5 M (FACS Buffer). Samples were run on a Cytoflex flow cytometer (Beckman Coulter, Life Sciences) and data were analyzed using FlowJo software.

### siRNA transfection

The short interfering RNAs (siRNAs) used in this study are the ON-Target plus human ATG5, ATG12, NBR1, SUSD6, TMEM127, and STUB1. Cells were transfected with siRNAs using Lipofectamine RNAiMax Transfection Reagent at a final concentration of 50 nM, according to the manufacturer's instructions.

### In vivo experiment

All animal experiments were approved by Ohio State University's Institutional Animal Care and Use Committee. CT26 ($1 \times 10^6$) cells were subcutaneously injected into the right flanks of Balb/c mice (8–10 weeks). Mice were monitored for tumor growth every 2 days afterward; tumor size was calculated using the formula (length × width × height)/2. When the tumors reached 500 mm³, mice were treated either with vehicle (saline) or papaverine (2 mg/kg) for 3 consecutive days via intravenous injection before tumor harvest.

### Tissue staining (frozen sections)

Slides were thawed at room temperature for 10 min. Sections were fixed with 1% paraformaldehyde for 10 min and permeabilized with 0.2% Triton X-100 in PBS, for 10 min, at room temperature. Slides were washed with 0.1% Tween-20 in PBS (PBS-T), and sections were then blocked with 10% normal goat serum in PBS-T for 45 min. Subsequently, sections were incubated in primary antibodies (GLUT1, CD8, and MHC I antibodies) diluted in blocking buffer overnight at 4 °C. Slides were washed in PBS-T, three times for 5 min. Sections were incubated with secondary antibodies for 45 min at room temperature in the dark. Following the incubation with secondary antibody, slides were washed in PBS-T, three times for 5 min. Sections were stained with Hoechst for 10 min at room temperature and washed three times with PBS 1×. Coverslips were mounted onto the slides using Prolong Gold mounting medium. The slides were imaged using Nikon-NiE.

### Multiplex staining and HALO analysis

Multiplex immunofluorescence (IF) staining was carried out on 4um thick formalin-fixed paraffin-embedded (FFPE) sections using the OPAL™ protocol (AKOYA Biosciences) on the Leica BOND RXm autostainer (Leica, Microsystems). Consecutive staining cycles were performed using primary Antibody-Opal fluorophore pairings. Primary antibodies were incubated for 30 min and detected using the BOND™ Polymer Refine Detection System (DS9800, Leica Biosystems) as per manufacturer's instructions, substituting the DAB for the Opal fluorophores, with a 10-min incubation time and without the hematoxylin step. Antigen retrieval at 100 °C for 20 min, as per standard Leica protocol, with Epitope Retrieval (ER) Solution 2 (AR9640, Leica Biosystems), was performed before each primary antibody was applied. For CA9, ER Solution 1 (AR9961, Leica Biosystems) was used. Sections were then incubated for 10 min with spectral DAPI (FP1490, Akoya Biosciences), and the slides mounted with VECTASHIELD® Vibrance™ Antifade Mounting Medium. Whole-slide scans and multispectral images (MSI) were obtained on the AKOYA Biosciences Vectra® Polaris™. Batch analysis of the MSIs from each case was performed with the inForm 2.4.11 software. Finally, batched analyzed MSIs were fused in HALO (Indica Labs), to produce a spectrally unmixed reconstructed whole tissue image, ready for analysis. HALO Image Analysis Platform version 3.6.4134.137 and HALO AI version 3.6.4134 (Indica Labs Inc.) were used. Analysis modules "Area Quantification 2.4.3", "High-Plex FL 4.1.3" and "Random Forest Classifier" were used.

### Gene expression correlations in TCGA datasets

RNA-sequencing data (RNA Seq V2 RSEM) for 595 colorectal adenocarcinoma patient sample datasets were extracted from the TCGA project, which can be accessed through CBIOPORTAL (http://www.cbioportal.org/). The hypoxia signature was determined by quantifying the median expression of genes from a validated hypoxic signature (Buffa et al, 2010). To examine the correlation of STUB1 and WWP2 against the hypoxia signature, Log10 conversion of the hypoxia signature was plotted against Log10 conversion of raw data for STUB1 and WWP2. Correlations and statistical significance were determined by calculating Spearman's rho rank correlation coefficients.

### Immunopeptidomics sample preparation

Frozen cell pellets were lysed in 3 ml lysis buffer (1% IGEPAL 630, 100 mM Tris pH 8.0, 300 mM NaCl supplemented with complete Protease Inhibitor Cocktail, EDTA-free, Roche) by pipetting gently up and down and incubating end-over-end at 4 °C. Lysates were then cleared by sequential centrifugation at 4 °C at first 500× g and then 21,000× g for 10 min and 1 h, respectively. Total of 3 mg of anti-MHC I monoclonal antibody (W6/32) or Hb-79 antibody conjugated to Protein A Sepharose using Dimethyl pimelimidate were added to the clarified supernatants and incubated with constant agitation overnight at 4 °C. Peptides MHC complexes were eluted with acetic acid and vacuum-dried. Resuspended MHC I eluted samples were centrifuged through 5kDA cutoff filters and cleared using Pierce C18 Spin Tips. Samples were vacuum-dried prior to LC-MS/MS acquisition.

### Lysate preparation

Frozen cell pellets were lysed in 3 ml lysis buffer (1% IGEPAL 630, 100 mM Tris pH 8.0, 300 mM NaCl supplemented with complete Protease Inhibitor Cocktail, EDTA-free, Roche) by pipetting gently up and down and incubating end-over-end at 4 °C. Lysates were then cleared by sequential centrifugation at 4 °C at first 500× g and then 21,000× g for 10 min and 1 h, respectively.

### Preparation of MHC class I immunoresin

MHC Class I antibody immunoresin for each biological replicate for HCT116 and HT29 cells was prepared by crosslinking 3 mg of antibody clone w6/32 to 0.5 ml of Sepharose protein A bead slurry in 10 column volumes (cv) of 40 mM dimethyl pimelimidate in borate buffer, pH 8.3 for 30 min at room temperature. The reaction was stopped with 10 cv of ice-cold 0.2 M Tris pH 8.0, followed by a washing step of 10 cv of 0.1% M citrate (pH 3.0) to remove unbound antibody, and the column was equilibrated with 10 cv of 50 mM Tris (pH 8.0). The same procedure was performed for the immunoaffinity capture of MHC molecules for the CT26 tumors samples clone 34.1.2 s (recognizing H2-Kd, Dd, and Ld, purified from hybridoma cells, ATCC HB-79).

### MS peptide enrichment and purification

Cleared lysates were incubated with immunoresin overnight at 4 °C under mild agitation. Columns were washed using 10 cv of 50 mM Tris pH 8.0 containing 150 mM NaCl, 450 mM NaCl, and a final wash with no salt. Peptide MHC complexes were eluted with the addition of 5 cv of 10% acetic acid. Samples were dried and resuspended in 0.1% TFA, 1% acetonitrile in water. Resuspended HLA-I-eluted samples were centrifuged through 5 kDa cutoff filters (Merck Millipore) and vacuum-dried. They were then resuspended in loading buffer and cleared using Pierce C18 Spin Tips (Thermo Fisher). Final elution was performed in 30% acetonitrile 0.1% TFA. Samples were vacuum-dried prior to LC-MS/MS acquisition.

### Proteomics sample preparation

Frozen pellets were lysed and cleared lysates were normalized to 15 μg per sample using the bicinchoninic acid Protein Assay (Peirce) and adjusted to 5% SDS in a final volume of 20 μl sample. Samples were reduced with 10 mM DTT for 15 min followed by alkylation with 55 mM iodoacetamide for a further 15 min and a repeated addition of 10 mM DTT. The reduced and alkylated

samples in 25 μl underwent digestion using the S-Trap midi protocol according to the manufacturer's instructions.

### LC-tandem mass spectrometry

MHC I peptides were dissolved in loading solvent (0.1% TFA, 1% Acetonitrile) and separated by an Ultimate 3000RSLCnano system (Thermo Scientific) using PepMAp C18 EASY-Spray LC column, 5-μm particle size, 75 μm × 50 cm column (Thermo Scientific) in 0.1% formic acid and coupled to a Q Exactive HF-X mass spectrometer (Thermo Scientific). Peptides were eluted with a gradient of either 2–25% for immunopeptidomics or 2–35% for proteomic samples in water containing 1% DMSO, 0.1% formic acid at a flow rate of 250 nl/min. For immunopeptidomics samples, data-dependent acquisition consisted of a 320–1600 $m/z$ full-MS scan (120,000 resolution, 100 ms accumulation time) and 20 dependent MS2 scans (60,000 resolution, 120 ms accumulation time). For proteomic samples, the data-dependent acquisition was adapted from a standard 15 Hz proteomics duty cycle 320–1600 $m/z$ full-MS scan (60,000 resolution, 45 ms accumulation time) and 12 dependent MS2 scans (30,000 resolution, 54 ms accumulation time). The quadruple isolation width was 1.6 $m/z$ (Immunopeptidomics) or 1.3 $m/z$ (Proteomics) and only 2–4 charge states were fragmented with a normalized high energy collisional dissociation energy set to 25% for immunopeptidomics and 28% for proteomic samples. Dynamic exclusion was set for 30 s, and all data were acquired in profile mode.

### Qualitative mass spectrometry data analysis

Mass spectrometry data were analyzed with Peaks v10.0 (Bioinformatics Solutions) for the identification of peptide sequences matching to databases (Human SwissProt protein entries (20391,027 Protein Entries, downloaded 19/10/2021) (Mouse SwissProt protein, downloaded 31/10/2022). Searches were performed with the following parameters: no enzyme specificity, no peptide modifications, peptide tolerance: ±5 ppm, and fragment tolerance: ±0.03 Da. The results were filtered using a peptide-level false discovery rate (FDR) of 1% established through parallel decoy database searches.

### Analysis of differential expression

For quantitative analysis, the data were analyzed by Progenesis QI for proteomics (Waters). Label-free quantification was performed through a calculation of area-based abundance on the top three unique peptides per protein. The method for normalization was scalar factor normalization to all proteins. All proteins which were quantifiable by progenesis were included for analysis. A two-way ANOVA analysis was applied to assess the significant regulation of peptides between the different conditions. Normalized protein quantification data was exported from progenesis, and differential expression analysis was performed using the Differential Expression of Proteins package (R version 4.2.0). Data imputation was performed using the maximum likelihood estimation method (maximum likelihood-based imputation). Volcano plots for each condition depicting −Log10 $P$ values (limma) against Log2 Fold Change showed significantly downregulated and upregulated proteins defined by a −Log10 $P$ cutoff of 5 and a Log2 Fold Change cutoff of 1. The same pipeline was performed for the quantitative analysis of immunopeptidomics data, except, label-free quantification was based on all associated peptides per protein.

### Pathways analysis

Pathway analysis was performed with Ingenuity Pathway Analysis (Qiagen). Differential expression data generated from the Differential Expression of Proteins analysis was inputted into the Ingenuity pipeline. A standard cutoff of 0.58 for LogFC which correlates to a FC of 1.5 in either direction and $P$ value < 0.05 was used for all analyses. The species was set to human for all analyses. Overrepresentation or 'expression' data used a LogFC of 0.58, which correlates to a FC of 1.5 only based on increased fold change.

### T-cell isolation and activation

Splenic murine CD8+ T lymphocytes were purified with negative CD8 microbeads by magnetic-activated sorting. Activation was done in complete RPMI supplemented with 25 μmol/mL 2-mercaptoethanol, 2 ng/mL recombinant mouse IL-2 and anti-mouse CD3/CD28. Purified OT-I CD8+T cells were activated with 1 μg/mL of the OVA-derived peptide SIINFEKL.

### In vitro cytotoxicity assay

In total, 10,000 MC38-Ova cells were seeded per well in 96-well plates (flat bottom, Costar) and treated under hypoxia (<0.1% O₂) for 16 h with Baf A1 and AMG PERK 44. After incubation, MC38-Ova were co-cultured for 6 h (<0.1% O₂) with mouse CD8+ OT-I cells. Wells were washed twice with PBS to remove T cells, and the number of remaining target cells was determined by culturing with Celltiter-blue for 1 h and measuring the fluorescence signal in a plate reader. Cytotoxicity was calculated relative to wells with no T cells added.

### Statistical analysis

Statistical analysis was carried out using GraphPad Prism Software. $P$ values were used to determine the significance of differences, and $P$ values of less than 0.05 were considered significant. Paired, Unpaired Student's $t$ test, one-way or two-way ANOVA, Fisher's exact test were used as appropriate and as specified in each figure legend.

## Data availability

The mass spectrometry proteomics and immunopeptidomics data have been deposited to the ProteomeXchange Consortium via the Pride partner repository with the data identifier PXD055205, PXD055207, PXD055211. The microscope images from Figs. 1F and 2H have been deposited in the BioImage Archive under accession number S-BIAD1349 https://www.ebi.ac.uk/biostudies/bioimages/studies/S-BIAD1349.

The source data of this paper are collected in the following database record: biostudies:S-SCDT-10_1038-S44318-024-00319-7.

## Peer review information

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

## Acknowledgements

This work was supported by Medical Research Council-UKRI grant MC_UU_00001/8, Cancer Research UK RadNet Centre Award C6078/A28736, NCI R01 CA255334, R01 CA262388, NIH grants P01CA275738/P01CA257804 and EPSRC EP/S019901/1. We would also like to acknowledge the expertise of the staff in the Microscopy Scientific Research Facility at the Department of Oncology, University of Oxford. We would also like to thank Matthew Jackson from Prof. Eileen Parkes' group and Vinnycius Pereira Almeida from Dr Carol Lung's group for providing OT-1 mice. Some of the figures were created using Biorender.com.

## Author contributions

**Hala Estephan**: Conceptualization; Resources; Formal analysis; Validation; Investigation; Visualization; Methodology; Writing—original draft; Writing—review and editing. **Arun Tailor**: Conceptualization; Formal analysis; Investigation; Visualization; Methodology; Writing—original draft. **Robert Parker**: Formal analysis; Methodology. **McKenzie Kreamer**: Investigation; Methodology. **Ioanna Papandreou**: Investigation; Methodology. **Leticia Campo**: Investigation; Methodology. **Alistair Easton**: Resources; Formal analysis. **Eui Jung Moon**: Validation. **Nicholas C Denko**: Resources; Methodology. **Nicola Ternette**: Validation; Project administration. **Ester M Hammond**: Conceptualization; Supervision; Writing—original draft; Project administration; Writing—review and editing. **Amato J Giaccia**: Conceptualization; Supervision; Funding acquisition; Writing—original draft; Project administration; Writing—review and editing.

Source data underlying figure panels in this paper may have individual authorship assigned. Where available, figure panel/source data authorship is listed in the following database record: biostudies:S-SCDT-10_1038-S44318-024-00319-7.

## Disclosure and competing interests statement

The authors declare no competing interests.

