## [Peer Review File · The EMBO Journal]

Hypoxia promotes tumor immune evasion by suppressing MHC-I expression and antigen presentation

Hala Estephan, Arun Tailor, Robert Parker, McKenzie Kreamer, Ioanna Papandreou, Leticia Campo, Alistair Easton, Eui Jung Moon, Nicholas C. Denko, Nicola Ternette, Ester M. Hammond, and Amato J. Giaccia

Corresponding author(s): Amato J. Giaccia (amato.giaccia@oncology.ox.ac.uk)

Review Timeline:

Submission Date:	4th Apr 24
Editorial Decision:	4th Jun 24
Revision Received:	4th Sep 24
Editorial Decision:	20th Sep 24
Revision Received:	29th Sep 24
Accepted:	14th Oct 24

Editor: Ioannis Papaioannou

Transaction Report:

Dear Dr. Giaccia,

Thank you for submitting your manuscript EMBOJ-2024-117498-T for consideration by The EMBO Journal, and for your patience during peer review. It has now been seen by two experts in the field, and we have received their reports, which you can find below.

As you will see, both referees are positive about the study and recognize the novelty and significance of the findings, as well as the proper design and quality of the work. In light of this input, I am glad to say that your manuscript would be suitable for publication in The EMBO Journal provided that a number of constructive concerns the referees raise and useful suggestions they make for strengthening the study and the manuscript further will be successfully addressed in a revision. In particular, we think that the last point (#3) of referee #2 (regarding the lack of data supporting the idea that preventing MHC-I reduction helps immune cells to target hypoxic cancer cells more effectively) would improve the manuscript significantly and thus encourage you to address it.

Given the referees' positive comments and recommendations, I would like to invite you to submit a revised version of the manuscript along with a detailed point-by-point response addressing all referees' comments. I should add that it is EMBO Journal policy to allow only a single round of major revision, and acceptance of your manuscript will therefore depend on the completeness of your responses in this revised version. Please let me know if you have any questions or comments that you would like to discuss with me.

We generally allow three months as standard revision time (September 3, 2024). As a matter of policy, competing manuscripts published during this period will not negatively impact our assessment of the conceptual advance presented by your study. However, we request that you contact us as soon as possible upon publication of any related work, to discuss how to proceed. Should you foresee a problem in meeting this three-month deadline, please let us know in advance and we may be able to grant an extension.

Thank you for the opportunity to consider your work for publication in The EMBO Journal. I look forward to your revision.

Best regards,

Ioannis

Instructions for preparing your revised manuscript

1. When you are ready to submit the revision, please upload:

- A Word file of the manuscript text (including legends of main Figures, EV Figures and Tables). Please make sure that changes are highlighted (or "tracked") to be clearly visible.

- Individual production-quality figure files (one file per figure). When assembling your figures, please refer to our figure preparation guidelines in order to ensure proper formatting and readability in print as well as on screen:

If the data shown in a figure are obtained from n {less than or equal to} 2, please use scatter plots showing the individual data points.

- i. the name of the statistical test used to generate error bars and P values
- ii. the number (n) of independent experiments (please specify technical or biological replicates) underlying each data point (discussion of statistical methodology can be reported in the Materials and Methods section, but figure legends should contain a basic description of n , P , and the test applied)
- iii. the nature of the bars and error bars (s.d., s.e.m.).

- A point-by-point response to the referees' comments, with a detailed description of the changes made (as a word file). All referees' concerns must be fully addressed and their suggestions taken on board. When preparing your letter of response to the referees' comments, please bear in mind that this will form part of the Review Process File and will therefore be available online to the community. Please note that you have the possibility to opt out of the transparent process at any stage prior to publication by letting the editorial office know (contact@embojournal.org); if you do opt out, the Review Process File link will point to the following statement: "No Review Process File is available with this article, as the authors have chosen not to make the review process public in this case.". For more details on our Transparent Editorial Process, please visit our website: <https://www.embopress.org/page/journal/14602075/authorguide#transparentprocess>

- Expanded View (EV) files (replacing Supplementary Information) that are collapsible/expandable online. A maximum of 5 EV Figures can be typeset. EV Figures should be cited as "Figure EV1, Figure EV2" etc. in the text, and their respective legends should be included in the manuscript file after the legends of regular figures. See detailed instructions regarding Expanded View files here: <https://www.embopress.org/page/journal/14602075/authorguide#expandedview>

- For the figures that you do NOT wish to display as Expanded View figures, they should be bundled together with their legends in a single PDF file called "Appendix", which should start with a short Table of Contents (including page numbers). Appendix figures should be referred to in the main text as: "Appendix Figure S1, Appendix Figure S2" etc. Please see detailed instructions here: <https://www.embopress.org/page/journal/14602075/authorguide#expandedview>

- A complete author checklist, which you can download from our author guidelines (<https://www.embopress.org/page/journal/14602075/authorguide>). Please note that the checklist will also be part of the Review Process File.

2. Please note that no statistics should be calculated and shown in Figures if $n=2$. Please also note that each p value should be reported as an exact value.

3. Before submitting your revision, primary datasets (and computer code, where appropriate) produced in this study need to be deposited in appropriate public databases (see <https://www.embopress.org/page/journal/14602075/authorguide#dataavailability>).

*** In particular, we kindly request you to submit your mass spectrometry data to an appropriate public database.***

The accession numbers and databases should be listed in a formal "Data availability" section (placed after Materials and Methods) that follows the model below (see also <https://www.embopress.org/page/journal/14602075/authorguide#dataavailability>):

Data availability

- RNA-seq data: Gene Expression Omnibus GSE46843 (<https://www.ncbi.nlm.nih.gov/geo/query/acc.cgi?acc=GSE46843>)
- [data type]: [name of the resource] [accession number/identifier/doi] ([URL or identifiers.org/DATABASE:ACCESSION])

*** All links should resolve to a page where the data can be accessed. ***

*** Please remember to provide in the Data availability section of your revised manuscript reviewer passwords if the datasets are not yet public. ***

*** The Data Availability Section is restricted to new primary data that are part of this study. In case you have no data that require deposition in a public database, please state so instead of referring to the database: "Our study includes no data deposited in public repositories." under the heading "Data availability". ***

4. Please check that the title and the abstract of the manuscript are brief, yet explicit, even to non-specialists. The length of the title should not exceed 100 characters, and the abstract should be a single paragraph not exceeding 175 words.

5. Please also note our reference format: <https://www.embopress.org/page/journal/14602075/authorguide#referencesformat>.

7. Please remember: digital image enhancement is acceptable practice, as long as it accurately represents the original data and conforms to community standards. If a figure has been subjected to significant electronic manipulation, this must be noted in the figure legend or in the "Materials and Methods" section. The editors reserve the right to request original versions of figures and

the original images that were used to assemble the figure.

8. Our journal encourages inclusion of data citations in the reference list to directly cite datasets that were obtained from public databases. Data citations in the article text are distinct from normal bibliographical citations and should directly link to the database records from which the data can be accessed. In the main text, data citations are formatted as follows: "Data ref: Smith et al, 2001" or "Data ref: NCBI Sequence Read Archive PRJNA342805, 2017". In the Reference list, data citations must be labeled with "[DATASET]". A data reference must provide the database name, accession number/identifiers, and a resolvable link to the landing page from which the data can be accessed at the end of the reference. Further instructions are available at: <https://www.embopress.org/page/journal/14602075/authorguide#referencesformat>.

9. We request authors to consider both actual and perceived competing interests. Please review our policy (<https://www.embopress.org/page/journal/14602075/authorguide#conflictsofinterest>) and update your competing interests statement if necessary. Please name this section 'Disclosure and competing interests statement' and place it after the Acknowledgements section.

10. Please note that all corresponding authors are required to provide an ORCID ID upon submission of a revised manuscript (<https://orcid.org/>). Please find instructions on how to link your ORCID ID to your account in our manuscript tracking system in our Author guidelines (<https://www.embopress.org/page/journal/14602075/authorguide#authorshipguidelines>).

11. We use CRediT to specify the contributions of each author in the journal submission system. CRediT replaces the author contribution section, which should be removed from the manuscript. Please use the free text box to provide more detailed descriptions. See also guide to authors: <https://www.embopress.org/page/journal/14602075/authorguide#authorshipguidelines>.

13. We would also welcome the submission of cover suggestions or motifs to be used by our Graphics Illustrator in designing a cover.

14. Please use the link below to submit your revision:
<https://emboj.msubmit.net/cgi-bin/main.plex>

Referee #1:

In this manuscript from Dr. Giaccia's group nicely investigated the role of hypoxia in regulating MHC-I responses. Tumor hypoxia in general, has been suggested to be a critical factor for tumor immunity, including MHC-I modulation (Yamamoto et al, 2020 Nature). However, the molecular mechanisms of which remains unclear. Combined with biochemical, imaging, experimental tumor models, the authors nicely examined a role of hypoxia in regulating MHC-I expression through PERK of the unfolded protein response and autophagy. This process also involves the newly identified MHC-I inhibitor SUSD6. They also performed immunopeptidomes-based LC-MS and found hypoxia reduced antigen profiles. Using autophagy inhibitors they found better MHC-I expression and immunopeptidome. These data are well-designed and provided interesting information to the field of tumor immunology, hypoxia, and MHC-I regulation.

I have several minor comments I see for potential improvement.

1. In figure 1, the authors examined the proximity of CD8 with hypoxia marker CAIX, which is very interesting... are there any controls for these experiments? For example, CD4 or other immune cells?
2. In figure 3, the authors found hypoxia induces SUSD6, how about WWP2 and other E3 ligase? Wondering the relative contribution of WWP2 mediated-ubiquitination versus UPR-PERK during hypoxia? Also, how about targeting SUSD6 and TMEM127 together in this case?
3. Figure 5 is very interesting, and important...it is better to use more cell lines to consolidate this conclusion. Also, what is the potential mechanism of hypoxia in affecting peptidomes? Could be helpful to provide more discussion. It would be wonderful if the authors can provide some data or discussion on the dominant versus subdominant epitopes in responsive to hypoxia, as this represents a critical question for immunotherapy (PMID:28939757; PMC5866601 etc.)

Referee #2:

The manuscript of Esthephan et al describes the effects of hypoxia on MHC-I expression and (partly) explains the immune evasive character of hypoxic cells. They elegantly show that this is mediated through autophagy targeting. As such, autophagy inhibition results in elevated MHC expression and increased antigen presentation. The manuscript is well written and conclusion are derived from the obtained results.

Hypoxia is a clinically relevant solid tumor phenomenon that severely limits effective therapies including immune therapy. Their findings could therefore be important for improving immune recognition of hypoxic cancer cells and as such improve immunotherapy outcome. Although I really appreciated the presented data, few issues would strengthen the manuscript significantly.

1) Although interesting and clinically meaningful, the administration of papaverine and the subsequent increase in T-cell invasion may not be related to reduction in hypoxia. For example, papaverine may have effects on T-cell migration and proliferation through e.g. CDK5 targeting or directly influence GLUT1 expression through insulin regulation. To robustly draw this conclusion the authors should at least quantify changes in hypoxia, but preferably target hypoxia by different means (e.g. hyperbaric oxygen, ARCON, or others).

2) Figure 4F would benefit from quantification. It is however surprising that autophagy-inhibition results in increased number of peptide expression as most of the MHC-I after baf1A seems to colocalize with LAMP1.

3) Although MHC-I mediated antigen expression is unmistakably important for immunogenicity, the largest caveat in the manuscript is, in my opinion, the lack of evidence that preventing MHC-I reduction aids cells of the immune system to recognize and kill hypoxic cancer cells more effectively. A syngeneic/HLA matched model with CT26 cells and mouse T-cells with autophagy inhibition and/or antibody mediated MHC blockade during hypoxia would really strengthen the clinical relevance of their findings.

Estephan et al., response to reviewers

Dear Dr Papaioannou,

Please see below for a detailed response to the reviewers' comments. Our responses are shown in blue text.

Referee #1:

In this manuscript from Dr Giaccia's group nicely investigated the role of hypoxia in regulating MHC-I responses. Tumor hypoxia in general, has been suggested to be a critical factor for tumor immunity, including MHC-I modulation (Yamamoto et al, 2020 Nature). However, the molecular mechanisms of which remains unclear. Combined with biochemical, imaging, experimental tumor models, the authors nicely examined a role of hypoxia in regulating MHC-I expression through PERK of the unfolded protein response and autophagy. This process also involves the newly identified MHC-I inhibitor SUSD6. They also performed immunopeptidomes-based LC-MS and found hypoxia reduced antigen profiles. Using autophagy inhibitors they found better MHC-I expression and immunopeptidome. These data are well-designed and provided interesting information to the field of tumor immunology, hypoxia, and MHC-I regulation.

1. In figure 1, the authors examined the proximity of CD8 with hypoxia marker CAIX, which is very interesting... are there any controls for these experiments? For example, CD4 or other immune cells?

We appreciate the reviewer's suggestion to include proximity analysis with other immune cells. In response, we have performed a proximity analysis of CD4⁺T cells and macrophages within colorectal cancer patient tissue microarrays and determined the percentage of CD4⁺T cells and macrophages in CAIX negative and positive areas. These new data are included in supplementary **Figure 1C-E** and the changes can be seen in the text lines 90-93.

2. In figure 3, the authors found hypoxia induces SUSD6, how about WWP2 and other E3 ligase? Wondering the relative contribution of WWP2 mediated-ubiquitination versus UPR-PERK during hypoxia? Also, how about targeting SUSD6 and TMEM127 together in this case?

Thank you for pointing this out. Our pathway analysis of the proteomics experiment in HT29 cells exposed to normoxia or hypoxia for 24 hours showed that the Protein Ubiquitination pathway was within the canonical pathways. The E3 ligase STUB1 was amongst the proteins

that appeared in this pathway. STUB1 has previously been found to be one of the negative regulators of MHC I expression (*Chen et al, Cell, 2023*). We investigated the role of STUB1 in regulating MHC I expression under hypoxic conditions. We found that the mRNA expression of STUB1 correlates positively with the hypoxia metagene signature indicating that STUB1 is a hypoxia inducible gene, in contrast WWP2 did not appear in our analysis and we did not find any positive correlation between the mRNA expression of WWP2 and the validated hypoxia signature. Most importantly, inhibition of STUB1 increased plasma membrane levels of MHC I in hypoxic HT29 cells supporting a role of this protein in regulating MHC I in hypoxia. These data have been included in the **Supplementary figure 4 J, K** and the respective changes have been incorporated in the manuscript (lines 179-187 and lines 354-359).

3. Figure 5 is very interesting, and important...it is better to use more cell lines to consolidate this conclusion. Also, what is the potential mechanism of hypoxia in affecting peptidomes? Could be helpful to provide more discussion. It would be wonderful if the authors can provide some data or discussion on the dominant versus subdominant epitopes in responsive to hypoxia, as this represents a critical question for immunotherapy (PMID:28939757; PMC5866601 etc.)

We acknowledge the reviewer comment, regarding the cell lines, we have observed the downregulation of the immunopeptidome in two colorectal cancer cell lines HCT116 (**Figure 5**) and HT29 (**Figure 6A-B**) in independent experiments. We would like to note here that the number of cells and cost to analyze the immunopeptidome in these experiments is quite high (e.g. minimum of 10^8 cells /condition/biological replicate). To conduct these analyses across many cell lines will require newer mass spectrometry technologies that will allow us to miniaturize these assays on a larger scale. Regarding the presence of dominant vs subdominant epitopes in response to hypoxia, we found that hypoxia inhibited all epitopes to a similar degree without bias to a certain epitope. The changes observed under hypoxia in the immunopeptidome were consistent suggesting no allele or length dependent effect (**Figure 5D, E**). In supplementary figure 6, we show some pathway level groupings among the downregulated epitopes being presented under hypoxia, but no specific pathways were significantly overrepresented. This indicates as per this dataset that hypoxia can downregulate the MHC I presentation in an unbiased manner.

Referee #2:

The manuscript of Estephan et al describes the effects of hypoxia on MHC-I expression and (partly) explains the immune evasive character of hypoxic cells. They elegantly show that this is mediated through autophagy targeting. As such, autophagy inhibition results in elevated MHC expression and increased antigen presentation. The manuscript is well written, and conclusion are derived from the obtained results.

Hypoxia is a clinically relevant solid tumor phenomenon that severely limits effective therapies including immune therapy. Their findings could therefore be important for improving immune recognition of hypoxic cancer cells and as such improve immunotherapy outcome. Although I really appreciated the presented data, few issues would strengthen the manuscript significantly.

Many thanks for this positive feedback

1) Although interesting and clinically meaningful, the administration of papaverine and the subsequent increase in T-cell invasion may not be related to reduction in hypoxia. For example, papaverine may have effects on T-cell migration and proliferation through e.g. CDK5 targeting or directly influence GLUT1 expression through insulin regulation. To robustly draw this conclusion the authors should at least quantify changes in hypoxia, but preferably target hypoxia by different means (e.g. hyperbaric oxygen, ARCON, or others).

2) Figure 4F would benefit from quantification. It is however surprising that autophagy-inhibition results in increased number of peptide expression as most of the MHC-I after baf1A seems to colocalize with LAMP1.

Thank you for pointing this out, the quantification of the colocalization between MHC I and LAMP1 has been added as **Figure 4 G**. We would also like to clarify that the increased number of peptides observed in the immunopeptidomics experiment after treating with Baf A1 under hypoxic conditions represents the total levels of MHC I including the intracellular level which are yet to be presented.

3) Although MHC-I mediated antigen expression is unmistakably important for immunogenicity, the largest caveat in the manuscript is, in my opinion, the lack of evidence that preventing MHC-I reduction aids cells of the immune system to recognize and kill hypoxic cancer cells more effectively. A syngeneic/HLA matched model with CT26 cells and mouse T-cells with autophagy inhibition and/or antibody mediated MHC blockade during hypoxia would really strengthen the clinical relevance of their findings.

We acknowledge the importance of demonstrating the role of hypoxia in modulating MHC I and peptide presentation *in vivo* and intend to make this the subject of a subsequent manuscript. To demonstrate functional characterisation, we used CD8⁺ OT-I T cells derived from mice carrying a transgene that possess a specific TCR that reacts to the SIINFEKL peptide of ovalbumin (OVA) that has been engineered to be expressed on MC38 colorectal tumor cells. MC38-Ova cells were exposed to hypoxia (<0.1% O₂) for 16 hours in the presence of Baf A1 or PERKi (AMG PERK44). Following this incubation, OT-I cells were co-cultured with MC38-Ova for 6 hours under normoxic or hypoxic (<0.1% O₂) conditions. An *in vitro* cytotoxicity assay was conducted to determine the increased antigen-specific cell killing against MC38-Ova in these conditions. The data showed a reduction in cell number when MC38-Ova were co-cultured with OT-I under hypoxia; this decrease in cell number was more pronounced when MC38-Ova were treated with Baf A1 and AMG PERK44, highlighting an enhancement of OT-I cell killing under these conditions in response to the prevention of MHC I degradation. These results provide evidence that preventing MHC I degradation under hypoxia by inhibiting either autophagy or the PERK arm of the UPR enhances the recognition of hypoxic cells by immune cells and improves the effectiveness of cell killing. These results have been incorporated in the manuscript **Figure 7F-G** and changes in the text have been applied (lines 312-321).

A. Scheme overview of the *in vitro* cytotoxicity assay workflow. B. Graph showing the percentage of cell viability for the indicated conditions. Data represents mean ± SEM from four independent biological replicates. Statistical analysis was calculated using 2way ANOVA test. ** $p < 0.005$, *** $p < 0.0001$.

We would like to emphasize that all change in the manuscript is highlighted in the revised version for ease of review. Once again, we thank the reviewers for their valuable feedback, which has significantly enhanced our work. We hope that the revisions are satisfactory, and that the manuscript is now suitable for publication in EMBO Journal.

Dear Prof. Giaccia,

Thank you for submitting your revised manuscript to The EMBO Journal. It has now been seen by the two original referees, and we have received their comments (included below). As you will see, both referees appreciate the additional work performed during revision, acknowledge that the previously raised concerns have been adequately addressed, and they now endorse your manuscript for publication in The EMBO Journal. There is only one minor comment from referee #1 (regarding the need for Figure 1 to be updated with the data on CD4/macrophages) that we would like you to address in a final version of your manuscript.

From the editorial side, there are also a few minor changes and corrections we need from you in the final version of your manuscript before we can proceed with its acceptance and publication:

- Please reduce the number of provided keywords to 5 (you currently list 7 keywords).
- Please note that the reference format must be updated: the names of the first 10 co-authors of each cited publication (followed by "et al." if necessary) must be listed for each reference. For more information on our reference format please visit: <https://www.embopress.org/page/journal/14602075/authorguide#referencesformat>.
- Thank you for providing referee access codes (tokens) for your deposited datasets. Now that review is complete, the referee access codes can be removed from the Data availability statement. Please make sure that all datasets will be publicly available at the time of publication, and provide the identifier and specific permanent URL for each dataset in a revised statement.
- Please correct "Disclosure and competing interest statement" to "Disclosure and competing interests statement".
- The author contributions statement should be removed from the manuscript file. Instead, we use CRediT to specify the contributions of each author in the journal submission system. Please feel free to use the free text box to provide more detailed descriptions during submission. See also our guide to authors for more information: <https://www.embopress.org/page/journal/14602075/authorguide#authorshipguidelines>.
- We noticed that callouts are missing for Fig. 5G-H, and Appendix Figure S9. Please make sure that all Figures and their panels are appropriately called out at least once in the main text.
- All Figures should be uploaded as individual, high-resolution files, and only their legends should remain in the main manuscript file below the References.
- Please move the information currently included in Appendix Table S1 (primer sequences) and Appendix Table S2 (antibodies) to your main Reagents and Tools table, and remove them from the Appendix (please also remember to update their callouts in the main text accordingly - and note that Appendix Table S1 was called out as "Appendix supplementary table 1").
- Please move the Supplementary methods from your Appendix to the Methods and Protocols section of your main manuscript file.
- Please remove the instructions from the top and the examples from the last page of your Reagents and Tools table.
- The Reagents and Tools table should only be uploaded as a separate file, please remove it from the main manuscript file.
- Please resize your synopsis image so that its width is exactly 550 pixels (the height is flexible, within the 300-600 pixels range). After resizing, please make sure that all text fonts are large enough for all text to be legible at the final image dimensions.
- During our standard image checks we detected possible cell re-use that needs to be explicitly mentioned in the figure legends:
 1. between Figure 1C and Appendix Figure S1A,
 2. between Figure 2F and Figure 5G.Please double-check these figure pairs, confirm that cells have been re-used, and -as long as this re-use is experimentally justified- please cite it in the respective figure legends.
- Please note that the legend of Figure 6g-h is mislabeled as 6g in the manuscript. This needs to be rectified.
- Please note that the exact p values are not provided in the legend of Figure 2g.
- Please indicate the statistical test used for data analysis in the legends of Figures 2a; 6g-h.
- Please note that the box plot needs to be defined in terms of minima, maxima, centre, bounds of box and whiskers, and

percentile in the legend of Figure 4g.

- Please note that information related to "n" is missing in the legends of Figures 1b; 5h.
- Please note that the error bars are not defined in the legend of Figure 1b.
- The order of the manuscript sections should be corrected: title page with complete author information, abstract, keywords, introduction, results, discussion, methods, data availability section, acknowledgements, disclosure and competing interests statement, references, main figure legends, tables, expanded figure legends.

Please also note that as part of the EMBO publications' Transparent Editorial Process, The EMBO Journal publishes online a Peer Review File along with each accepted manuscript. This File will be published in conjunction with your paper and will include the referee reports, your point-by-point response and all pertinent correspondence relating to the manuscript. You can opt out of this by letting the editorial office know (contact@embojournal.org). If you do opt out, the Peer Review File link will point to the following statement: "No Peer Review File is available with this article, as the authors have chosen not to make the review process public in this case."

We look forward to seeing a final version of your manuscript as soon as possible. Please let us know if you have any questions and use this link to submit your revision: <https://emboj.msubmit.net/cgi-bin/main.plex>

Best wishes,

Ioannis

Referee #1:

I appreciate the helpful responses from the authors. In general, they reasonably answered my questions. However, I did not find the data on CD4/macrophages in Figure 1. The authors should update such data.

Referee #2:

Immune therapy is the latest significant and novel improvement in cancer care. Hypoxia limits its efficacy. The significance of this manuscript is therefore high. It also provides a potential novel therapy that overcomes these limitations. As such it is not only scientifically important but may have future societal impact too. After the first round of revision, the authors elaborately addressed all the issues that were raised and provided novel important evidence for their conclusions. As such, I do not have any further concerns.

All editorial and formatting issues were resolved by the authors.

Dear Amato,

Congratulations on an excellent manuscript! I am very pleased to inform you that it has been accepted for publication in The EMBO Journal. Thank you very much for comprehensively addressing the concerns initially raised by the referees and for making the requested editorial changes.

Your manuscript will be processed for publication by EMBO Press. There are a few remaining minor changes needed, about which I will contact you again within the next few days. Your manuscript will then be copy edited and you will receive page proofs prior to publication. Please note that you will be contacted by Springer Nature Author Services to complete licensing and payment information.

If you have any questions, please do not hesitate to contact the Editorial Office. Thank you for your contribution to The EMBO Journal. Working with you has been a pleasure!

Best regards,

Ioannis
